

# Mechanistic Insights into I₂O₅ Heterogeneous Hydrolysis and Its Role in Iodine Aerosol Growth in Pristine and Polluted Atmospheres

Xiucong Deng[1], An Ning[1,*], Ling Liu[1], Fengyang Bai[1,2], Jie Yang[1], Jing Li[1], Jiarong Liu[1], and Xiuhui Zhang[1,*]

[1]State Key Laboratory of Environment Characteristics and Effects for Near-space, Key Laboratory of Cluster Science, Ministry of Education of China, School of Chemistry and Chemical Engineering, Beijing Institute of Technology, Beijing, 100081, China
[2]Institute of Catalysis for Energy and Environment, College of Chemistry and Chemical Engineering, Shenyang Normal University, Shenyang, 110034, China

*Correspondence to*: A. Ning (anning@bit.edu.cn) and X. H. Zhang (zhangxiuhui@bit.edu.cn)

**Abstract.** Higher iodine oxides are intricately linked to marine aerosol formation; however, the underlying physicochemical mechanisms remain poorly constrained, particularly for $I_2O_5$, which is stable yet conspicuously absent in the atmosphere. While reactivity with water has been implicated, the direct hydrolysis of $I_2O_5$ ($I_2O_5 + H_2O \rightarrow 2HIO_3$) fails to account for this discrepancy due to its high activation barrier (21.8 kcal mol⁻¹). Herein, we have probed heterogeneous hydrolysis of $I_2O_5$

mediated by prevalent chemicals over oceans through Born-Oppenheimer molecular dynamics and well-tempered metadynamics simulations. Our results demonstrate that self-catalyzed pathways involving $I_2O_5$ and its hydrolysis product $HIO_3$ substantially reduce the reaction barrier, thereby accelerating the conversion of $I_2O_5$ to $HIO_3$ in pristine marine environments. In polluted regions, interfacial hydrolysis of $I_2O_5$ mediated by acidic or basic pollutants (e.g., $H_2SO_4$ or amines) proceeds with even greater efficiency, characterized by remarkably low barriers (≤1.3 kcal mol⁻¹). Collectively,

these proposed heterogeneous reactions of $I_2O_5$ are highly effective, acting as a hitherto unrecognized sink for $I_2O_5$ and a source of $HIO_3$—processes that facilitate marine aerosol growth and rationalize the high iodate abundances detected in aerosols. These findings provide mechanistic insight into the elusive $I_2O_5$-to-$HIO_3$ conversion, offering a critical step toward improving the representation of iodine chemistry and marine aerosol formation in atmospheric models, with implications for climate prediction and environmental impact assessment.

## 1 Introduction

Given the vast coverage (71%) of oceans on Earth's surface, marine aerosols exert substantial impact on the global atmosphere, altering cloud microphysical properties, radiative balance, and climate change (Kaloshin, 2021; Mahowald et al., 2018). Hence, understanding how aerosols form is fundamental for the climate system, especially ruinous extreme weather, as it introduces the greatest uncertainty in climate forcing (Gettelman and Kahn, 2025). Earlier studies have confirmed a

solid association between marine aerosols and sulfur chemicals derived from the oxidation of dimethyl sulfide (DMS) (Barnes et al., n.d.; Chen et al., 2018). While the recent evidence has shown that iodine-bearing precursors of oceanic origin





such as iodine oxides ($I_2O_{2-5}$) and iodine oxyacids ($HIO_{2-3}$) significantly contribute to marine aerosol formation, owing to their high efficiency and rising levels (Allan et al., 2015; Carpenter et al., 2021; Droste et al., 2021; Gómez Martín et al., 2021, 2022b; Huang et al., 2022; Li et al., 2022; McFiggans et al., 2010; O'Dowd et al., 2002; O'Dowd and De Leeuw, 2007; Roscoe et al., 2015; Saiz-Lopez et al., 2012, 2014). Despite the experimental and theoretical studies uncovering the role of $HIO_{2-3}$ in aerosol formation (He et al., 2021, 2023; Liu et al., 2023; Zhang et al., 2022, 2024; Zu et al., 2024), the atmospheric fate and impacts of $I_2O_{2-5}$ are yet to be fully established (Gómez Martín et al., 2013, 2020; Leroy and Bosland, 2023; Lewis et al., 2020; Liang et al., 2022), limiting the accuracy of atmospheric models in simulating iodine-driven climate effects.

As a typical higher $I_2O_{2-5}$, iodine pentoxide ($I_2O_5$) serves as the anhydride of iodic acid ($HIO_3$), playing a vital role in bridging iodine oxides and iodic acid in iodine chemical cycle, however, its true fate is still elusive, with the conflicting findings reported. Specifically, $I_2O_5$ is highly thermally stable (Kaltsoyannis and Plane, 2008), yet it is not evidently present in the gas phase, implying its unknown sinks. Sipilä et al. (2016) hypothesized that $I_2O_5$ may participate in the formation of marine aerosol particles through the reaction (R1: $2HIO_3 \rightarrow I_2O_5 + H_2O$) based on the identified oxygen-to-iodine ratios of 2.4 in field-collected particles. Nevertheless, no direct detection of $I_2O_5$, such as mass spectral signals, was provided. And the theoretical study has shown that reaction R1 can hardly occur due to its high energy barrier (27.8 kcal mol$^{-1}$) and endothermic nature (Khanniche et al., 2016), suggesting that this sink pathway seems not well-supported. More recently, the experimental evidence indicates that $I_2O_5$ can indeed participate in aerosol particle formation, but under the unrealistically low-humidity conditions (Rörup et al., 2024). Upon the addition of water, $I_2O_5$ is markedly depleted, suggesting its reaction with water, accompanied by the formation of $HIO_3$ (Gómez Martín et al., 2022a). This finding seems to support well the absence of $I_2O_5$ in high-humidity marine atmospheres. While the theoretical study of Xia et al. (2020) found the direct reaction between $I_2O_5$ and $H_2O$ (R2: $I_2O_5 + H_2O \rightarrow 2HIO_3$) in gas phase also needs to cross a high energy barrier of 21.8 kcal mol$^{-1}$, suggesting that this process is unlikely to occur effectively, even with another $H_2O$ or $I_2O_5$ acting as a catalyst (Kumar et al., 2018). As a result, this gas-phase mechanism alone appears insufficient to explain the observed absence of $I_2O_5$ in experimental or clean marine environment, with the $I_2O_5$ sink remaining unrevealed. Furthermore, Kumar et al. (2018) proposed the heterogeneous reactions of $I_2O_5$ at the air-water interface, which is ubiquitous over oceans like aerosol surface (Zhong et al., 2018). Unfortunately, this interfacial mechanism still fails to account for the observed well-established link between $I_2O_5$ depletion and $HIO_3$ formation, as it points to the formation of other novel product $H_2I_2O_6$. Despite its dynamic stability within the simulation scale (34 ps) (Kumar et al., 2018), $H_2I_2O_6$ has not been detected in the prior iodine-related experimental and field studies (Gómez Martín et al., 2020, 2022a; He et al., 2021, 2023; Rörup et al., 2024; Sipilä et al., 2016), highlighting the poor understanding of the reaction mechanism of $I_2O_5$ under pristine conditions.

The mystery of $I_2O_5$ chemistry likely deepens in polluted coastal regions, e.g., Zhejiang, China (Yu et al., 2019), where the evident burst of iodine aerosols has been also observed. Similarly, $I_2O_5$ is also evidently absent but $HIO_3$ abundant in the field-collected aerosols, indicating $I_2O_5$ may undergo an $I_2O_5$-to-$HIO_3$ conversion. Yet the underlying mechanisms in polluted regions remain unclear, which is likely more complex than that in pristine regions due to the pollutant-mediated





impacts from typical amine or sulfuric acid that can facilitate the heterogeneous reactions of other iodine oxides (Ning et al., 2023, 2024). Taken together, these unresolved discrepancies and mechanistic uncertainties render the fate of $I_2O_5$ elusive in both pristine and polluted marine environment, which limits our understanding of its true role in atmospheric iodine chemistry and aerosol formation.

In this study, we have performed the Born-Oppenheimer molecular dynamics (BOMD) simulations to elucidate the potential hydrolysis mechanism of $I_2O_5$ at the air-water interface, as represented by aqueous aerosol surfaces, mediated by different kinds of prevalent chemicals. We first investigate the heterogenous hydrolysis of $I_2O_5$ mediated by $I_2O_5$ and $HIO_3$, which potentially occur in pristine marine regions; while the reactions of $I_2O_5$ mediated by pollutants such as acid (sulfuric acid, SA) and alkaline (methylamine (MA), dimethylamine (DMA), trimethylamine (TMA)) were also probed to

comprehend the loss path of $I_2O_5$ in polluted marine environment. Alongside mechanistic insights, we further quantified the reaction barriers by metadynamics simulations (MetaD) and block error analysis. By comparing the resulting energy barriers, the dominant mechanisms of $I_2O_5$ hydrolysis under different environment were identified. Furthermore, we performed wavefunction analysis to examine the reactive sites of $I_2O_5$ hydrolysis products at the air-water interface, which may facilitate the condensation of gaseous precursors to promote aerosol growth. This work provides mechanistic insights into

heterogeneous hydrolysis of $I_2O_5$ that potentially occur under marine conditions, enhancing our understanding of iodine chemistry and marine aerosol formation.

## 2 Methods

### 2.1 Quantum Chemistry Calculations

Structural optimizations for gas-phase molecular geometries were performed using the Gaussian 16 software package (Frisch

et al., 2016) with tight convergence criteria. During the geometry optimization, the M06-2X functional was selected for gas-phase molecules due to its excellent performance in calculating main group elements (Zhao and Truhlar, 2008), and the adopted basis set was aug-cc-pVTZ (Kendall et al., 1992) (for C, H, O, N, and S atoms) + aug-cc-pVTZ-PP with ECP28MDF (for I atom) (Peterson et al., 2003). The frequency analysis was performed at the same level of theory to ensure that the optimized structures have no imaginary frequencies.

### 2.2 Classical MD Simulations

Classical molecular dynamics (MD) simulations were carried out by the GROMACS 2022 package (Van Der Spoel et al., 2005) to probe the interfacial propensity of $I_2O_5$ at air-water interface. Regarding the initial configurations for MD simulations, we used the PACKMOL code (Martínez et al., 2009) to build a cubic water slab with 3.3 nm edge, consisting simulation box of 1000 water molecules with a tolerance of 2.0 Å (minimum interatomic distance). And the simulation box

was expanded to $3.3 \times 3.3 \times 16$ $nm^3$ along the *Z*-axis to build the vacuum layers, forming the air-water interface. Umbrella sampling was employed, and a harmonic bias of 5000 kJ $mol^{-1}$ $nm^{-2}$ was applied to $I_2O_5$ molecule along the *Z*-axis to



investigate its free energy profile crossing the air-water interface from bulk water into gas phase. Furthermore, the generalized AMBER force field (GAFF) (Wang et al., 2004) with parameters obtained from the Sobtop 1.0 code (Lu, 2024) was combined with the TIP4P water model (Jorgensen et al., 1983) to describe the driving forces in the MD simulations. The restrained electrostatic potential (RESP) method (Bayly et al., 1993) was used to fit the atomic charges using the Multiwfn 3.7 code (Lu and Chen, 2012). Prior to MD simulations, the energy minimization of the system was performed by the steepest descent algorithm. The MD simulations were carried out in the constant volume and temperature ($NVT$) ensemble, with the temperature controlled at 300 K by the Nosé−Hoover thermostat (Evans and Holian, 1985). A time step of 2.0 fs was employed for all MD simulations. Lennard−Jones (LJ) and Coulomb potentials were used to simulate nonbonded interactions, with the electrostatic interactions calculated using the particle mesh Ewald (PME) summation method (Darden et al., 1993) and a 10 Å real-space cutoff for nonbonded interactions. The hydrogen-involved bonds were treated via the LINCS algorithm (Hess et al., 1997). In each umbrella sampling window, the system was equilibrated for 5 ns. The free-energy profile was calculated by the weighted histogram analysis method (WHAM) (Kumar et al., 1992).

**2.3 BOMD Simulations**

To explore the interfacial reaction of $I_2O_5$, the underlying mechanism was studied by BOMD and the stepwise multi-subphase space metadynamics (SMS-MetaD) (Fang et al., 2022) simulations using the CP2K 2022 program (Kühne et al., 2020) combined with PLUMED software (Anon, 2019). The QUICKSTEP module was applied with the Gaussian and plane wave (GPW) method, and the electronic exchange−correlation term was described by the BLYP-D3 functional (Perdew and Wang, 1992). The DZVP-MOLOPT-SR-GTH basis set and Goedecker−Teter−Hutter (GTH) pseudopotentials (Goedecker et al., 1996; Hartwigsen et al., 1998) were adopted here. The plane-wave cutoff was set to 280 Ry, and that for Gaussian was 40 Ry. All BOMD simulations were performed in the $NVT$ ensemble, with the temperature controlled at 300 K by the Nosé−Hoover thermostat (Evans and Holian, 1985), and the time step was set to 1.0 fs. The dimensions of 3-D periodic cell are $x = 18$ Å, $y = 18$ Å, and $z = 30$ Å, in which the water slab containing 128 $H_2O$ was fully relaxed to reach a statistical equilibrium for temperature and potential energy (see Figure S1 in SI). To probe the reaction barriers of $I_2O_5$ hydrolysis mediated by different chemicals, a series of SMS-MetaD simulations were carried out. In the SMS-MetaD simulations, collective variable (CV), as the key parameter, was set to effectively differentiate distinct states (Figures S5 and S6), with its upper and lower limits referring to the interfacial behavior observed in unbiased simulations (see more details in SI). Gaussian hills with adaptive heights and sigma widths of 0.1 Å were deposited every 50 steps to efficiently explore the free energy landscape and accelerate the convergence (Figures S7-S10). And block average analysis was performed to assess the error associated with the calculated reaction barriers (Figure S11). The wave function analysis and visualization of key structures from the BOMD simulations were carried out using VMD 1.9.3 (Humphrey et al., 1996) and Multiwfn 3.7 software (Lu and Chen, 2012).





## 3 Results and Discussion

### 3.1 Surface Preference

The interfacial preference of $I_2O_5$ influences whether its heterogeneous reactions can effectively occur. Herein, umbrella sampling was applied to obtain the free energy profile of $I_2O_5$ from the bulk water to the air along the $Z$ direction, where $I_2O_5$ was initially placed in the center of water layer. As shown in Figure 1, the air-water interface is defined as the region with $10\% - 90\%$ of the bulk density, delineated by the two dashed lines. As $I_2O_5$ moves from the bulk water into air across the interface, the free energy first decreases and then increases, reaching the lowest values near the Gibbs dividing surface (GDS,

the blue dash line). The results show that $I_2O_5$ has an evident preference for aqueous surface, suggesting its potential to react with other chemicals at the air-water interface.

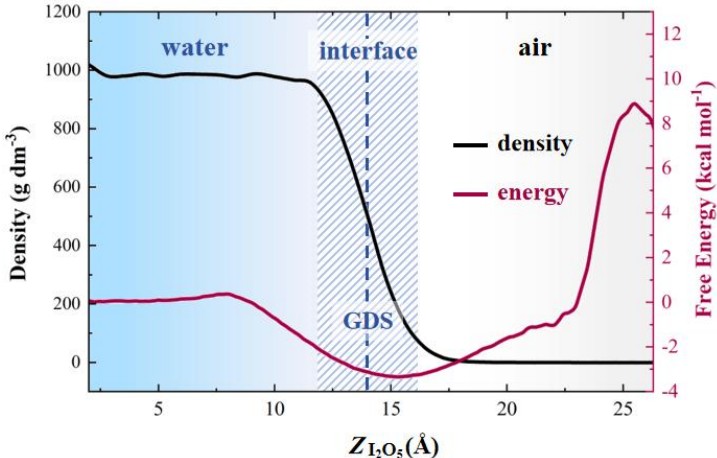

**Figure 1.** Interfacial propensity of $I_2O_5$. Mass density (g dm$^{-3}$) of the water (black line); the free energy of $I_2O_5$ (purple line) along the $Z$

axis. The vertical dashed line indicates the Gibbs dividing surface (GDS), which corresponds to a location with half of the bulk density. The shaded region represents the air-water interface ($10\% - 90\%$ of bulk density).

### 3.2 H$_2$O-Mediated Interfacial Hydrolysis of I$_2$O$_5$

Experimental evidence (Gómez Martín et al., 2022a) has confirmed a high reactivity of $I_2O_5$ in the presence of $H_2O$, which stands in contrast to theoretical predictions based on gas-phase reactions (Khanniche et al., 2016; Xia et al., 2020). To

address this gap, we first probed unexplored heterogeneous hydrolysis of $I_2O_5$ at the air-water interface by the BOMD simulations. The results indicate no apparent tendency for $I_2O_5$ to react with interfacial $H_2O$ to form $HIO_3$ in the unbiased BOMD simulation within 50 ps (Figure S4), implying this process may involve an energy barrier. Accordingly, the enhanced sampling (i.e., SMS-MetaD) simulations were further performed to examine the interfacial reactions of $I_2O_5$ and to quantify the associated reaction barrier (see CV settings in SI). Figure 2(a)-(b) illustrates the process of $I_2O_5$ hydrolysis, where the

reaction initiates with proton (H$^+$) transfer from $H_2O$ to $I_2O_5$ facilitated by another $H_2O$ as a 'bridge'. Then $I_2O_5$ binds with



$H^+$ to form the intermediate $HI_2O_5^+$, with the system existing in a quasi-transition state (TS). Subsequently the I–O bond in the $HI_2O_5^+$ breaks, and the $OIO^+$ motif quickly combines with the $OH^-$ to form a $HIO_3$, leaving another $HIO_3$. Figure 2(c) presents the free energy profile for the interfacial hydrolysis of $I_2O_5$, showing a reaction barrier of ~9.4 kcal $mol^{-1}$ upon convergence of the results (Figure S7), with $\pm 0.1$ kcal $mol^{-1}$ error as indicated by the block analysis (see more details in SI),

which is lower than that of the reported corresponding gas-phase reaction (13.6 kcal $mol^{-1}$) (Xia et al., 2020). However, this barrier remains considerable, limiting the direct heterogeneous hydrolysis from proceeding rapidly. The previous study (Kumar et al., 2018) reported the formation of $H_2I_2O_6$ from the reaction of $I_2O_5$ with interfacial water and indicated its dynamic stability over a 35 ps simulation period. However, $H_2I_2O_6$ was unfortunately not observed in the reported iodine-related experimental and field studies (Gómez Martín et al., 2020, 2022a; He et al., 2021, 2023; Rörup et al., 2024; Sipilä et

al., 2016), nor was it observed in our unbiased BOMD simulations. Thus, $H_2O$-mediated heterogeneous mechanism still faces challenges in representing the identified $I_2O_5$-to-$HIO_3$ conversion, pointing to the faster pathways mediated by other chemicals responsible for the absence of $I_2O_5$ in the experimental or field observations.

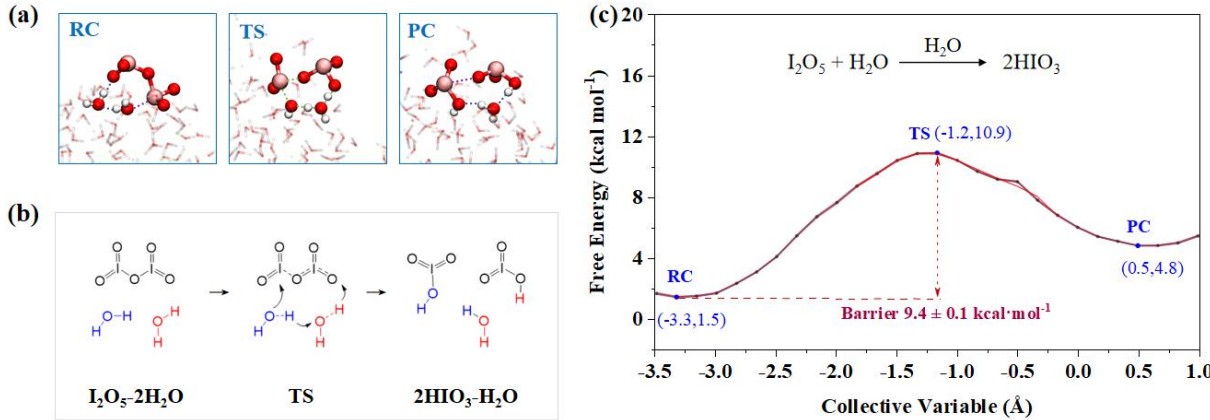

**Figure 2.** Details of $H_2O$-mediated hydrolysis of $I_2O_5$. (a) Snapshot structures of SMS-MetaD simulation representing reactant complex (RC), transition state (TS), and product complex (PC). Pink, red, white atoms represent I, O, H in sequence (The same below). (b) Reaction mechanism for the $H_2O$-mediated hydrolysis of $I_2O_5$. Blue molecular indicates the water involved in hydrolysis, black one indicates the $I_2O_5$, red one indicates the catalyst. The arrows indicate the direction of atom transfer (The same below). (c) Free energy profile for $I_2O_5$ hydrolysis at the air-water interface mediated by $H_2O$ (black line) with the error bands (pink region).

**3.3 Self-Mediated Hydrolysis of $I_2O_5$**

Given that $H_2O$-mediated hydrolysis is inefficient, we next explored whether leftover $I_2O_5$ can self-catalyze a faster reaction—a critical question for explaining its atmospheric depletion. During the unbiased BOMD simulation, two $I_2O_5$ molecules combined together to form iodine oxide complex ($I_2O_5\cdots I_2O_5$) via intermolecular halogen bond (XB), without undergoing hydrolysis within 50 ps (Figure S4). Further SMS-MetaD simulations, shown in Figure 3(a)-(b), present the




mechanistic details of this hydrolysis. The $I_2O_5 \cdots I_2O_5$ complex initially binds a $H_2O$ molecule via hydrogen bond (HB) and XB, forming a three-membered ring. Next, the $H_2O$ molecule dissociates into $OH^-$ and $H^+$, each approaching a distinct $I_2O_5$. Meanwhile, the $H_2O$-involved HB (O–H$\cdots$O) and XB (O–I$\cdots$O) generally converted into new H–O and I–O covalent bonds, resulting in the formation of two $HIO_3$ molecules. And the residual OIO and $IO_3$ of the two $I_2O_5$ molecules combined to form a new $I_2O_5$, finalizing the self-catalyzed reaction. As shown in Figure 3(c), the resulting reaction barrier is 3.5 kcal mol$^{-1}$,

which is 5.9 kcal mol$^{-1}$ lower than that of $H_2O$-catalyzed mechanism. The results suggest that $I_2O_5$ facilitates its own hydrolysis more pronouncedly than $H_2O$. Thus, the $I_2O_5$-mediated mechanism efficiently drives $I_2O_5$-to-$HIO_3$ conversion, resolving the earlier paradox of $I_2O_5$'s stability yet absence in the atmosphere—and explaining the high $HIO_3$ levels in marine aerosols.

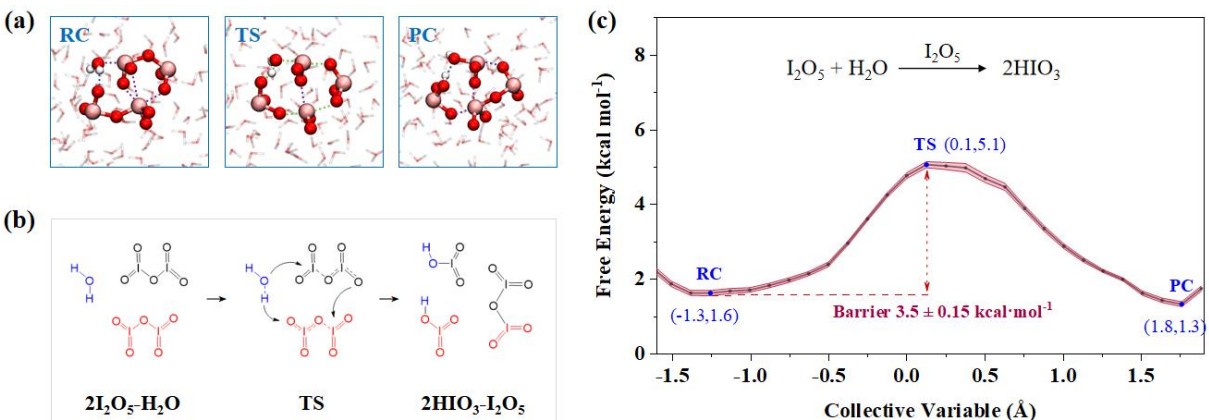


**Figure 3.** Details of $I_2O_5$-mediated hydrolysis of $I_2O_5$. (a) Snapshot structures of SMS-MetaD simulation representing RC, TS, and PC. (b) Reaction mechanism for the $I_2O_5$-mediated hydrolysis of $I_2O_5$. (c) Free energy profile for $I_2O_5$ hydrolysis at the air-water interface mediated by $I_2O_5$ (black line) with the error bands (pink region).

### 3.4 HIO$_3$-Mediated Interfacial Hydrolysis of I$_2$O$_5$

We further performed SMS-MetaD simulations to explore the interfacial reaction mediated by the product $HIO_3$, due to its widespread presence in gas and aerosol phase. And $HIO_3$ possesses both proton donor and acceptor sites, potentially facilitating proton transfer, which is a key step in $I_2O_5$ hydrolysis. As shown in Figure 4(a)-(b), $H^+$ transfers from $H_2O$ to $HIO_3$, then $HIO_3$ donates $H^+$ to $I_2O_5$, finally yielding two new $HIO_3$. The resulting energy barrier of this process is ~4.3 kcal mol$^{-1}$ (Figure 4(c)), slightly higher than that mediated by $I_2O_5$ (3.5 kcal mol$^{-1}$). However, given the high concentration and

widespread distribution of $HIO_3$ over oceans, its role in mediating hydrolysis of $I_2O_5$ should be fully considered. Moreover, as this reaction proceeds, the product $HIO_3$ accumulates to further catalyze the reaction, establishing positive feedback. Thus, in the pristine iodine-rich region such as Mace Head, with $HIO_3$ concentrations as high as $10^8$ molec. cm$^{-3}$, (Sipilä et al., 2016) the $HIO_3$-catalyzed heterogeneous hydrolysis of $I_2O_5$ may represent its primary loss pathway. This may help explain



why $I_2O_5$, though hypothesized to be involved in aerosol formation, is not observed in mass spectra, where iodate signals
appear instead.

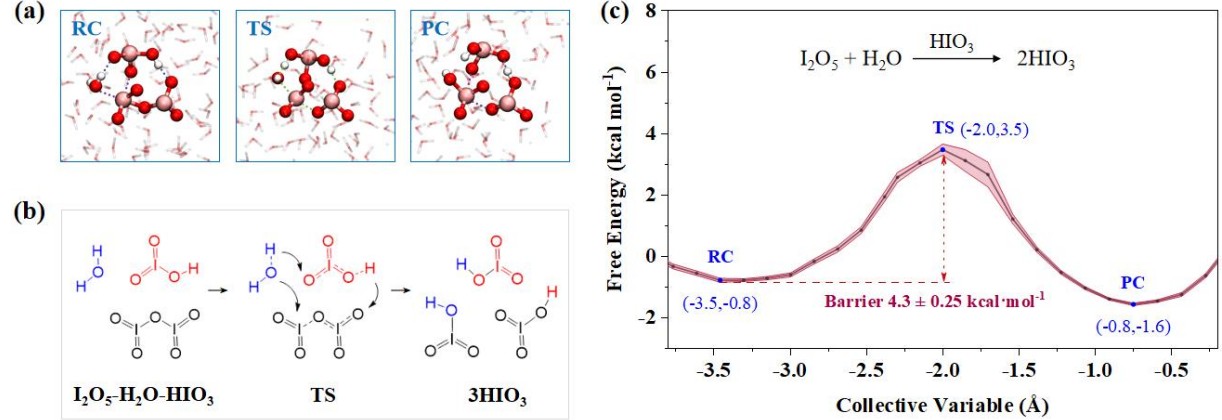

**Figure 4.** Details of HIO$_3$-mediated hydrolysis of I$_2$O$_5$. (a) Snapshot structures of SMS-MetaD simulation representing RC, TS, and PC. (b)
Reaction mechanism for the HIO$_3$-mediated hydrolysis of I$_2$O$_5$. (c) Free energy profiles for I$_2$O$_5$ hydrolysis at the air-water interface
mediated by HIO$_3$ (black line) with the error bands (pink region).

**3.5 Sulfuric Acid-Mediated Interfacial Hydrolysis of I$_2$O$_5$**

In fact, I$_2$O$_5$ seems not only absent over pristine oceans but also in polluted marine environment. For example, along the
eastern coast of China (e.g., Zhejiang), bursts of iodine-bearing aerosols were observed, featuring high levels of IO$_3^-$, without
I$_2$O$_5$ signal detected (Yu et al., 2019). Despite the proposed conversion of hydrated I$_2$O$_5$ to HIO$_3$ (detected as IO$_3^-$) (Gómez
Martín et al., 2022a), the underlying mechanism remains unclear, especially under the impacts of pollutants. Notably, HSO$_4^-$
(ionization of typical acidic pollutant H$_2$SO$_4$) was also detected at considerable concentrations within iodine aerosols.
Accordingly, we explored the H$_2$SO$_4$-mediated hydrolysis of I$_2$O$_5$ at the molecular level. As a strong acid (pK$_{a1}$ = -3.0),
H$_2$SO$_4$ can rapidly lose a proton to form HSO$_4^-$ at the air-water interface, which has a surface preference (Hua et al., 2015).
At this stage, HSO$_4^-$ loses strong acidity and shows dissociation equilibrium, thus serving as a proton-transfer bridge. As
shown in Figure 5(b), H$^+$ initially moves from HSO$_4^-$ to I$_2$O$_5$, meanwhile, the H atom of H$_2$O is attracted by SO$_4^{2-}$ and
remains the OH$^-$, leading to I–O bond breaking, forming two HIO$_3$ molecules. This process couples the self-dissociation of
HSO$_4^-$ and the hydrolysis of I$_2$O$_5$, expected to occur efficiently on aqueous surface, owing to its lower barrier (~1.3 kcal mol$^{-1}$) than that mediated by I$_2$O$_5$ (3.5 kcal mol$^{-1}$). The findings indicate that HSO$_4^-$-mediated pathway may play an important role
in I$_2$O$_5$ depletion under polluted conditions, thereby providing mechanistic insights into the absence of I$_2$O$_5$ and the
coexistence of HSO$_4^-$ and IO$_3^-$ in field-collected aerosols (Yu et al., 2019). To further quantify the efficiency of these
reactions, we calculated the reaction rate constants through Transition State Theory (Table S1). Notably, the hydrolysis of
I$_2$O$_5$ mediated by I$_2$O$_5$ or HIO$_3$ proceeds 4–5 orders of magnitude faster than that mediated by H$_2$O, but still more slowly (1-2



orders of magnitude) than that mediated by $H_2SO_4$. These results highlight the significance of intercomponent coupling in enhancing heterogeneous iodine chemistry in marine atmospheres, from pristine to polluted.

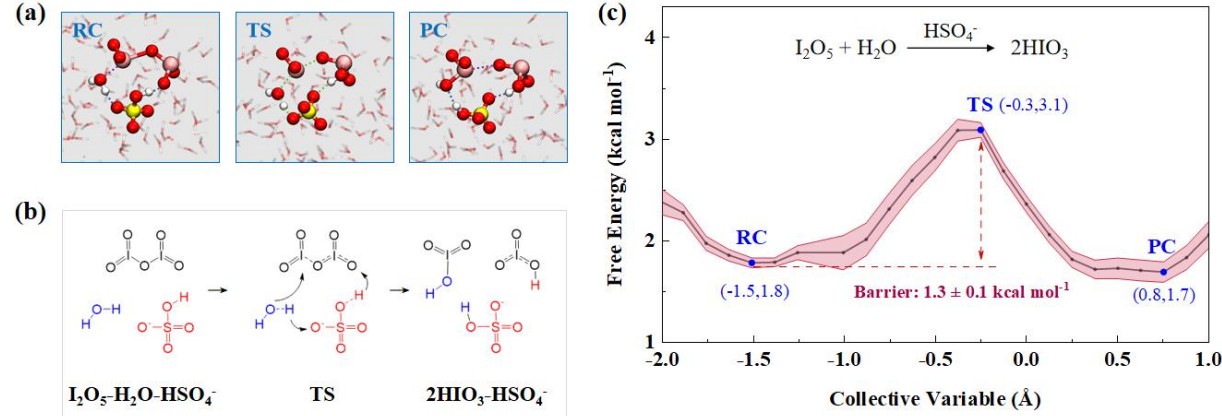

**Figure 5.** Details of SA-mediated hydrolysis of $I_2O_5$. (a) Snapshot structures of SMS-MetaD simulation representing RC, TS, and PC. Yellow atoms represent S. (b) Reaction mechanism for the SA-mediated hydrolysis of $I_2O_5$. (c) Free energy profiles for $I_2O_5$ hydrolysis at the air-water interface mediated by SA (black line) with the error bands (pink region).

**3.6 Amine-Mediated Interfacial Hydrolysis of $I_2O_5$**

Aside from acid pollutants, atmospheric bases may also affect interfacial chemistry under polluted conditions. Amines, as the typical bases, exhibit evident surface preference and the potential to affect hydrolysis of iodine oxides at the air-water interface (Ning et al., 2024). Accordingly, we investigated the amine-mediated hydrolysis of $I_2O_5$. Figure 6 shows the DMA-involved key structural snapshots and time-dependent evolution of key bond distances. Initially, DMA is positioned in the gas phase, without notable interaction with aqueous surface. Over time, DMA gradually approaches interfacial water to form HB (O3–H1···N). At 1.12 ps, interfacial water bridges $I_2O_5$ and DMA via HB (O3–H1···N) and XB (O2–I2···O3), forming a pre-reaction complex. By 1.19 ps, the covalent bond between the H1 and O3 elongated, forming a TS-like structure, then proton H1 transfers from $H_2O$ to DMA. The deprotonated $H_2O$ molecule, generating $OH^-$, immediately approaches the I2 atom of $I_2O_5$. The I–O bond in $I_2O_5$ gradually changes from covalent bond into XB, ultimately yielding $HIO_3$ and $IO_3^-$ (Figure S15). By 1.23 ps, the reaction ends to form $HIO_3$, $IO_3^-$, and alkylammonium salts ($DMAH^+$). The BOMD results indicate that the DMA-mediated hydrolysis of $I_2O_5$ occurs rapidly on a picosecond timescale, thus attesting to a barrierless pathway (Fang et al., 2024). Also, we studied the MA- and TMA-mediated hydrolysis of $I_2O_5$, which shows similar capabilities of other amines like MA and TMA to accelerate this process (see Figures S2 and S3 in SI). Accordingly, in polluted environments, the role of pollutants in activating iodine chemistry should not be overlooked, as their mediation of rapid heterogeneous processes likely affects aerosol composition and growth dynamics.



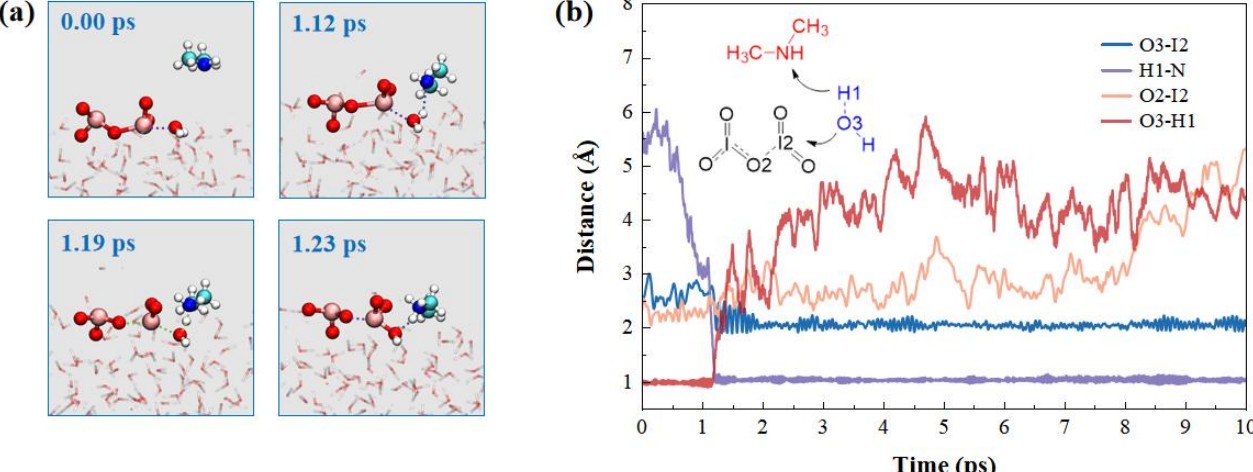

**Figure 6.** Details of DMA-mediated hydrolysis of $I_2O_5$. (a) Snapshot structures of BOMD simulation, which illustrate the mechanism for this process. The dashed lines indicate the intermolecular interactions (hydrogen or halogen bonding). Blue and cyan atoms represent N and C atoms, respectively. (b) Time evolution of key bond distances (O3–I2, H1–N, O2–I2, and O3–H1). The arrows indicate the direction of atom transfer.

In addition to the above analysis of the reaction mechanism and energy barriers, we further explored how interfacial hydrolysis of $I_2O_5$ affects aerosol growth (Scheme 1) through analysis of dynamic trajectories and wave function of key structures. Here, the DMA-mediated process is taken as an example (see details in SI). The resulting products ($HIO_3$, $IO_3^-$, and $DMAH^+$) are partially solvated, as they are surrounded by interfacial water (Figures S12 and S13) via HBs and XBs but still exposed outside to air with unoccupied HB or XB sites (Figure S14). These HB and XB interactions serve the following roles: i) stabilizing the product complex by preventing its decomposition via intermolecular interactions within complex; ii) inhibiting evaporation of product via interactions with interfacial water; iii) promoting aerosol growth by unoccupied HB or XB sites for uptake of gaseous species, in particular, the hydrophilic ionic products (e.g., $MAH^+$, $DMAH^+$, and $TMAH^+$) accelerate water condensation. Thus, the chemisorption of $I_2O_5$ contributes to aerosol growth not only directly, by yielding low-volatility products, but also indirectly by enhancing hygroscopic growth. Understanding the impacts of these heterogeneous reactions and the resulting products may advance our knowledge of how iodine oxides influence iodate aerosol formation.





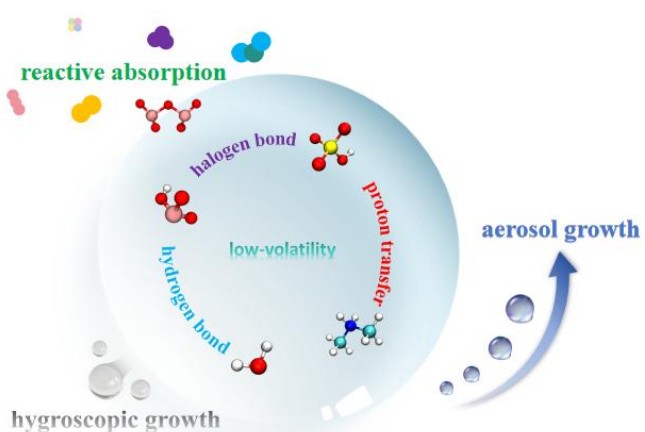

**Scheme 1.** Illustration of the potential mechanism of aerosol growth by $I_2O_5$ hydrolysis. The representative products of $I_2O_5$ hydrolysis are shown as the same forms mentioned above. The gathered balls with various colors indicate the prevalent gaseous molecules in the atmosphere.

## 265  4 Conclusion

Iodine chemicals are closely linked to marine aerosol formation; however, the underlying physicochemical process, especially involving higher iodine oxides (e.g., $I_2O_x$, $x \geq 2$), remains highly uncertain. $HIO_3$ is widely detected in marine aerosols, while $I_2O_5$ is absent, pointing to a potential $I_2O_5$-to-$HIO_3$ conversion, although the mechanism is still elusive. To address this, we have investigated interfacial hydrolysis of $I_2O_5$, the typical $I_2O_x$, on aqueous aerosol surfaces at the

molecular level by BOMD and SMS-MetaD simulations. The results show that $I_2O_5$ uptakes directly onto the pure water surface, where the interfacial hydrolysis proceeds slowly ($I_2O_5 + H_2O \xrightarrow{H_2O} HIO_3$ in Figure 2). And the self-catalyzed pathways involving $I_2O_5$ (reactant) and $HIO_3$ (product) notably lower the energy barrier and facilitate the reaction. Thus, in pristine marine environments, the proposed $I_2O_5$- and $HIO_3$-mediated heterogeneous mechanisms (Figures 3 and 4) potentially serve as both an important sink for $I_2O_5$ and a source of $HIO_3$, which helps clarify the chemical speciation of

field-collected iodine aerosols. In contrast, in polluted regions, pollutant-mediated pathways—with barriers as low as 1.3 kcal mol$^{-1}$ ($H_2SO_4$-mediated) or even barrierless (amine-mediated)—dominate over self-catalyzed processes, driving faster $I_2O_5$ interfacial hydrolysis and yielding $HIO_3$, which helps to elucidate the rapid formation of iodate aerosols observed in polluted eastern coast of China (Yu et al., 2019). It also highlights the role of air pollutants in activating iodine chemistry. Accordingly, these heterogeneous mechanisms can advance our understanding of the roles of $I_2O_5$ in iodine chemistry, which

may help explain the well-established but mechanistically elusive $I_2O_5$-to-$HIO_3$ conversion. Notably, the hydrolysis products (i.e. $HIO_3$, $IO_3^-$, and ionic species) enhance aerosol growth through their low volatility, unoccupied HB/XB sites, and hygroscopicity—linking $I_2O_5$ hydrolysis directly to aerosol growth. These insights deepen our understanding of how iodine oxide affects aerosol formation—both directly and indirectly—in both pristine and polluted marine environments.





Altogether, this study highlights the critical role of heterogeneous hydrolysis of $I_2O_5$ mediated by some important
chemicals in iodine chemistry and aerosol growth. Here, we find that intercomponent coupling-i.e., interactions of iodine
oxides with themselves, iodic acids, and atmospheric pollutants-is a key, yet frequently overlooked factor that can reshape
heterogenous iodine chemistry across environments, where the reactive pollutant-mediated processes outcompete self-
catalyzed pathways. These findings can help explain the observed but poorly understood $I_2O_5$ and $HIO_3$ distribution, as well
as their chemical conversion, which sheds light on iodine aerosol growth. The iodine-driven impacts are likely global and
expected to grow, due to rapidly increasing oceanic iodine emissions. Accordingly, integrating these proposed mechanisms
into atmospheric models is expected to reduce uncertainties (unrecognized sink or source for iodine oxides and iodine
oxyacids) arising from iodine oxides, which can refine the representation of iodine cycling and further improve the
assessment of globally environmental and climatic effects (like aerosol-driven radiative forcing) driven by atmospheric
iodine aerosols.



**Data availability.** The data in this article are available from the corresponding author upon reasonable request (anning@bit.edu.cn and zhangxiuhui@bit.edu.cn).

**Supplement.** The supplement related to this article is available online at :

**Author contribution.** XZ designed and supervised the research. XD and AN performed the quantum chemical calculations and the BOMD simulations. XD, AN, JL and LL analyzed data. XD, AN, and XZ wrote the paper. FB, JY, and JRL reviewed the paper. All authors commented on the paper.

**Competing interests.** The contact author has declared that neither they nor their co-authors have any competing interests.

**Disclaimer.** Publisher's note: Copernicus Publications remains neutral with regard to jurisdictional claims in published maps and institutional affiliations.

**Acknowledgement.** We acknowledge financial support from the National Science Fund for Distinguished Young Scholars and the National Natural Science Foundation of China.

**Financial support.** This work is supported by the National Science Fund for Distinguished Young Scholars (grant no. 22225607) and the National Natural Science Foundation of China (grant no. 22306011 and 42105101).

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
