# Peer review of "Mechanistic Insights into $I_2O_5$ Heterogeneous Hydrolysis and Its Role in Iodine Aerosol Growth in Pristine and Polluted Atmospheres"

_EGUsphere, 2025_

## Author Comment (AC1)

**Responses to Referee #1's comments**

We are grateful to the reviewers for their professional and helpful comments on our manuscript "Mechanistic Insights into I2O5 Heterogeneous Hydrolysis and Its Role in Iodine Aerosol Growth in Pristine and Polluted Atmospheres" (MS No.: egusphere-2025-3770). Accordingly, we have carefully revised the manuscript. The point-to-point responses to the Referee #1's comments are summarized below:

Deng et al. presented a theoretical study showing that, under the influence of atmospheric iodine species and pollutants, I2O5 hydrolysis can occur more readily at the surface of aqueous aerosols. These physicochemical processes are valuable, as I2O5, being a key chemical in the iodine cycling, has a significant impact on both iodine chemistry and the formation of iodine aerosols. Experimental investigation of gas—liquid interfacial reactions is challenging; therefore, the heterogeneous mechanisms revealed by the authors through Ab initio molecular dynamics simulations provide an important advancement of the previous understanding of the atmospheric fate of iodine oxides. This manuscript is thoughtfully prepared, with reliable methods and comprehensive data in both the main text and the supplementary material that support the conclusions. That said, certain aspects could benefit from minor revision, and I recommend publication after the authors have addressed my comments.

**Response:** Thanks for the reviewer's professional and valuable comments. We have addressed all comments point by point and made the corresponding revisions in the manuscript. The detailed responses are listed as follows.
* * *
**Major Comment:**

Page 3, lines 84-85: It is stated that gas-phase structure optimizations were performed with Gaussian package, yet the species involved are not clearly identified. As I could not find this information in either the manuscript or the SI. If I have overlooked it, please direct me to the relevant section.

**Response**: We thank the reviewer for the careful review. This suggestion has reminded us of the missing details in the manuscript. In this study, we optimized the gas-phase conformations of the reactants (e.g., I2O5, HIO3, and H2SO4) using the Gaussian 16 program (Frisch et al.,

2016). Accordingly, we have added the relevant molecular information in the Methods section in the revised manuscript (Page 3, line 92), as follows:

"The geometries and coordinates of gas-phase molecules (i.e. I2O5, HIO3, H2SO4, MA, DMA, and TMA) are provided in Figure S2 and Table S2 in the supporting information (SI), respectively." in the Quantum Chemistry Calculations section.

For ease of review, we have copied them as follows:

**Figure S2.** The optimized structures for gas-phase molecules (i.e. I2O5, HIO3, H2SO4, methylamine (MA), dimethylamine (DMA), and trimethylamine (TMA)) at the M06-2X//aug-cc-pVTZ(-PP) level of theory.

**Table S2.** Coordinates for all calculated molecules at the M06-2X/aug-cc-pVTZ(-PP) level of theory.

|   | I         | 2 O 5 (Isomer 1) |           |
|---|-----------|----------------------------------------|-----------|
| О | -0.000013 | -0.988788                              | 0.000159  |
| I | 1.641204  | 0.055287                               | -0.225857 |
| I | -1.641235 | 0.055311                               | 0.225852  |
| О | 2.800249  | -1.108887                              | 0.404898  |
| О | 1.270721  | 1.236952                               | 1.040258  |
| О | -2.800236 | -1.108859                              | -0.404989 |
| О | -1.270515 | 1.236874                               | -1.040296 |

|   |           | HIO 3               |           |
|---|-----------|--------------------------------|-----------|
| I | -0.096935 | 0.013334                       | -0.244580 |
| O | -0.289743 | 1.559142                       | 0.594271  |
| O | -1.014657 | -1.204758                      | 0.647080  |
| O | 1.713186  | -0.419126                      | 0.233680  |
| Н | 1.867280  | -0.188774                      | 1.162521  |
|   |           | H 2 SO 4 |           |
| S | 0.000001  | -0.000003                      | 0.154681  |
| O | 0.670149  | 1.064615                       | 0.826194  |
| O | 1.023747  | -0.686964                      | -0.845067 |
| Н | 1.708834  | -0.047610                      | -1.086390 |
| O | -1.023744 | 0.686990                       | -0.845052 |
| Н | -1.708821 | 0.047636                       | -1.086403 |
| О | -0.670155 | -1.064638                      | 0.826162  |
|   |           | MA                             |           |
| N | 0.747530  | 0.000000                       | -0.120845 |
| Н | 1.149839  | -0.811832                      | 0.328231  |
| Н | 1.149839  | 0.811832                       | 0.328231  |
| C | -0.706161 | 0.000000                       | 0.017785  |
| Н | -1.112738 | 0.876800                       | -0.483829 |
| Н | -1.112739 | -0.876799                      | -0.483831 |
| Н | -1.069944 | -0.000001                      | 1.050407  |
|   |           | DMA                            |           |
| N | 0.000000  | 0.568475                       | -0.148304 |

| Н | 0.000001  | 1.336588  | 0.508868  |
|---|-----------|-----------|-----------|
| C | -1.204665 | -0.224024 | 0.020309  |
| Н | -2.083499 | 0.413833  | -0.056533 |
| Н | -1.258202 | -0.965087 | -0.778566 |
| Н | -1.244160 | -0.762560 | 0.977876  |
| C | 1.204665  | -0.224024 | 0.020309  |
| Н | 1.258206  | -0.965080 | -0.778573 |
| Н | 2.083499  | 0.413834  | -0.056523 |
| Н | 1.244156  | -0.762568 | 0.977871  |
|   |           | TMA       |           |
| N | 0.000000  | 0.000009  | -0.389205 |
| C | 1.195212  | 0.683133  | 0.062542  |
| Н | 2.079355  | 0.170403  | -0.315039 |
| Н | 1.202277  | 1.705136  | -0.315089 |
| Н | 1.260883  | 0.720681  | 1.163047  |
| C | -1.189213 | 0.693506  | 0.062541  |
| Н | -1.187224 | 1.715614  | -0.314877 |
| Н | -2.077799 | 0.188704  | -0.315260 |
| Н | -1.254710 | 0.731401  | 1.163041  |
| C | -0.005990 | -1.376629 | 0.062538  |
| Н | -0.892218 | -1.885891 | -0.314819 |
| Н | 0.875409  | -1.893854 | -0.315333 |
| Н | -0.006029 | -1.452314 | 1.163040  |
* * *
For the central chemical examined in this work, the I2O5 molecule, different isomers are expected to exist. Could the authors clarify why the current structure was selected and on what basis? Moreover, in the introduction the I2O5 molecule is described as very stable; does the cited reference pertain to the same structure investigated here? The rationale for the chosen structure should be stated, and the atomic coordinates together with a structural figure are best included in the SI.

Response: The reviewer's comment is professional. In our manuscript, the isomers of I2O5 had been already considered before carrying out BOMD simulations. We have surveyed the previously reported structures of I2O5 (Kaltsoyannis and Plane, 2008; Khanniche et al., 2016; Kim and Yoo, 2016). Although isomer 2 (Kaltsoyannis and Plane, 2008) is referred to in the Introduction as a stable configuration, our calculations show that isomer 1 (Khanniche et al., 2016) is in fact more stable, with a lower Gibbs free energy. Thus, the most stable isomer 1 was selected for the subsequent BOMD simulations. We have supplemented description for isomers of I2O5 in the Methods section as follows (Page 3, line 86):

"The I2O5 molecule with lowest Gibbs free energy has been selected from isomers (Kaltsoyannis and Plane, 2008; Khanniche et al., 2016; Kim and Yoo, 2016), and details of the structures and coordinates are provided in SI (Figure S1 and Table S3)."

To ensure clarity for the readers, in the revised SI, we have presented the considered isomers together with their calculated Gibbs free energies in Fig. S1, and the corresponding coordinates are summarized in Table S3.

**Figure S1.** The optimized structures for isomers of I2O5 at the M06-2X/aug-cc-pVTZ(-PP) level of theory. The relative Gibbs free energies (kcal/mol, comparing to isomer 1) are provided beneath the corresponding isomers.

**Table S3.** Coordinates for all calculated isomers of  $I_2O_5$  at the M06-2X/aug-cc-pVTZ(-PP) level of theory.

| icory.   |           |           |           |  |  |
|----------|-----------|-----------|-----------|--|--|
|          |           | Isomer 1  |           |  |  |
| O        | -0.000013 | -0.988788 | 0.000159  |  |  |
| I        | 1.641204  | 0.055287  | -0.225857 |  |  |
| I        | -1.641235 | 0.055311  | 0.225852  |  |  |
| О        | 2.800249  | -1.108887 | 0.404898  |  |  |
| О        | 1.270721  | 1.236952  | 1.040258  |  |  |
| О        | -2.800236 | -1.108859 | -0.404989 |  |  |
| О        | -1.270515 | 1.236874  | -1.040296 |  |  |
| Isomer 2 |           |           |           |  |  |
| I        | 0.000000  | 1.656872  | -0.054736 |  |  |
| О        | 0.000000  | 0.000000  | 0.989073  |  |  |
| I        | 0.000000  | -1.656872 | -0.054736 |  |  |
| О        | -0.789433 | -2.716528 | 1.107431  |  |  |
| О        | 1.199506  | 1.113881  | -1.239338 |  |  |
| О        | 0.789433  | 2.716528  | 1.107431  |  |  |
| О        | -1.199506 | -1.113881 | -1.239338 |  |  |
| Isomer 3 |           |           |           |  |  |
| I        | -1.769146 | -0.160199 | -0.157389 |  |  |
| O        | -0.000002 | 0.000007  | -0.980462 |  |  |
| I        | 1.769147  | 0.160192  | -0.157396 |  |  |
| O        | -2.140622 | 1.528856  | 0.182916  |  |  |
| O        | -1.400607 | -0.996020 | 1.350040  |  |  |
| О        | 1.400657  | 0.996070  | 1.350017  |  |  |
* * *
Across the heterogeneous hydrolysis pathways of I2O5 presented in this study, whether mediated by water, iodic acid, I2O5, or pollutants, the cleavage always occurs at the central I–O covalent bond of the I2O5. This appears to be a consequence of the CV definition, which biases the system toward iodic acid formation. Nevertheless, the rationale for this setup should be substantiated by chemical evidence. A wavefunction analysis, such as bond order calculations, could be provided to confirm that the central I–O bond is indeed the weak, thereby justifying its designation as the most likely bond to break.

Response: According to the reviewer's suggestion, we have calculated the Mayer bond order (MBO) of adopted I2O5 molecule by Multiwfn 3.7 (Lu and Chen, 2012). As shown in Figure S3, the central I-O bond is considered to be a single bond (MBO: 0.864), while the terminal I-O bond is thought to be a double bond (MBO: 1.729). The results indicate that initial cleavage is expected to occur at the central but weaker I-O covalent bond of the I2O5. We have supplemented the chemical-bond characterization for this part in the SI as the reviewer suggested. This result provides compelling chemical evidence supporting the rationality of the CV settings.

**Figure S3.** Mayer bond orders for I2O5 calculated at the M06-2X/cc-pVTZ(-PP) level of theory.
* * *
Atmospheric iodine species and pollutants are more diverse than the limited set examined here. For example, even in the case of amines, more than one hundred species exist in the atmosphere. It would be helpful if the authors could include a brief discussion of the possible roles of other atmospheric components, or at minimum acknowledge this as a limitation of the current study.

Response: Thank you for this insightful suggestion. This helps readers understand the limitations of the study. The real atmosphere is complex; as the reviewer noted, iodine species (e.g. HOI, HIO2, HIO3, I2O3, I2O4, and I2O5) and atmospheric pollutants (e.g. H2SO4, HNO3, dimethyl sulfide, organic acids, and aromatic hydrocarbons)-including amines-are highly diverse. We consider that other components are also likely to influence the heterogeneous hydrolysis process of the I2O5 of interest. In this study, the effects of I2O5 and HIO3 on the reaction are explored here mainly because, as reactants and products, they are most likely to coexist in the same environment, thereby facilitating self catalysis. Meanwhile, we chose H2SO4 and amines (i.e. MA, DMA, and TMA) as the representative acid and base pollutants that are associated with aerosol particle formation. We have expanded this part of the discussion to better reflect real atmospheric conditions and acknowledge the limitation that the manuscript can not comprehensively examine all species in revised manuscript as follows (Page 12, line 304):

"The real atmosphere is chemically complex, including iodine species (e.g. HOI, HIO2, HIO3, I2O3, I2O4, and I2O5) and atmospheric pollutants (e.g. H2SO4, HNO3, organic acids, and ammonia), which are likely to influence the heterogeneous hydrolysis of I2O5. In future work, we intend to confirm the impacts from other atmospheric components."
* * *
**Minor Comments:**

Page 5, Line 139: "...along the Z axis..." Units are missing.

**Page6, Line 165**: "Pink, red, white atoms represent I, O, H in sequence (The same below)." This sentence should be: "The pink, red, and white spheres represent I, O, and H atoms, respectively (the same applies in Figures 3–6 below)."

Lines 203 and 226: 'Profiles' should be 'The profile'

Lines 188: 'error bands' should be 'error band'

**Line 240**: It is recommended to remove this citation, as it does not appear to provide effective support.

**Response**: We appreciate the reviewer's careful evaluation. Accordingly, we have completed all corresponding revisions in response to the reviewer's minor comments.
* * *
**Reference:**

- Frisch, M. J.; Trucks, G. W.; Schlegel, H. B.; Scuseria, G. E.; Robb, M. A.; Cheeseman, J. R.;
  Scalmani, G.; Barone, V.; Petersson, G. A.; Nakatsuji, H.; Li, X.; Caricato, M.;
  Marenich, A. V.; Bloino, J.; Janesko, B. G.; Gomperts, R.; Mennucci, B.; Hratchian,
  H. P.; Ortiz, J. V.; Izmaylov, A. F.; Sonnenberg, J. L.; Ding, F.; Lipparini, F.; Egidi,
  F.; Goings, J.; Peng, B.; Petrone, A.; Henderson, T.; Ranasinghe, D.; Zakrzewski, V.
  G.; Gao, J.; Rega, N.; Zheng, G.; Liang, W.; Hada, M.; Ehara, M.; Toyota, K.;
  Fukuda, R.; Hasegawa, J.; Ishida, M.; Nakajima, T.; Honda, Y.; Kitao, O.; Nakai, H.;
  Vreven, T.; Throssell, K.; Montgomery, J. A., Jr.; Peralta, J. E.; Ogliaro, F.;
  Bearpark, M. J.; Heyd, J. J.; Brothers, E. N.; Kudin, K. N.; Staroverov, V. N.; Keith,
  T. A.; Kobayashi, R.; Normand, J.; Raghavachari, K.; Rendell, A. P.; Burant, J. C.;
  Iyengar, S. S.; Tomasi, J.; Cossi, M.; Millam, J. M.; Klene, M.; Adamo, C.; Cammi,
  R.; Ochterski, J. W.; Martin, R. L.; Morokuma, K.; Farkas, O.; Foresman, J. B.; Fox,
  D. J. Gaussian 16 Rev. A.01; Gaussian, Inc.: Wallingford, CT, 2016.
- Kaltsoyannis, N. and Plane, J. M. C.: Quantum chemical calculations on a selection of iodine-containing species (IO, OIO, INO3, (IO)2, I2O3, I2O4 and I2O5) of importance in the atmosphere, Phys. Chem. Chem. Phys., 10, 1723, https://doi.org/10.1039/b715687c, 2008.
- Khanniche, S., Louis, F., Cantrel, L., and Černušák, I.: Computational study of the I2O5+ H2O = 2 HOIO2 gas-phase reaction, Chem. Phys. Lett., 662, 114–119, https://doi.org/10.1016/j.cplett.2016.09.023, 2016.
- Kim, M. and Yoo, C.-S.: Phase transitions in I2O5 at high pressures: Raman and X-ray diffraction studies, Chem. Phys. Lett., 648, 13–18, https://doi.org/10.1016/j.cplett.2016.01.043, 2016.
- Lu, T. and Chen, F.: Multiwfn: A multifunctional wavefunction analyzer, J. Comput. Chem., 33, 580–592, https://doi.org/10.1002/jcc.22885, 2012.

---

## Author Comment (AC2)

**Responses to Referee #2's comments**

We are grateful to the reviewers for their professional and helpful comments on our manuscript "Mechanistic Insights into I2O5 Heterogeneous Hydrolysis and Its Role in Iodine Aerosol Growth in Pristine and Polluted Atmospheres" (MS No.: egusphere-2025-3770). Accordingly, we have carefully revised the manuscript. The point-to-point responses to the Referee #2's comments are summarized below:

The manuscript by Deng et al. adopted first-principles molecular dynamics to examine the heterogeneous hydrolysis of  $I_2O_5$  and its role in aerosol growth. The study identifies interfacial mechanisms driven by iodic species in pristine conditions and by acid/base pollutants in polluted environments. These findings emphasize the importance of reactive atmospheric components in  $I_2O_5$  hydrolysis and provide a mechanistic explanation for its sink and the observed  $I_2O_5$ -to-HIO3 conversion. This topic is timely and relevant to Atmos. Chem. Phys., given its focus on aerosol formation process. The manuscript is overall well presented, with sound methodology and adequate supporting evidence. Nevertheless, I suggest that the authors consider my comments and perform the minor revisions before the manuscript can be recommended for publication.

**Response**: We appreciate the insightful and constructive suggestions. According to these comments, we have responded and revised the manuscript as follows.
* * *
1. In conducting the metadynamics simulations, the authors appear to have employed SMS-MetaD rather than the more widely used MetaD approach. I am curious about the rationale behind this choice. The authors should explain the advantages of this method in the Methods section. Furthermore, to substantiate its feasibility, some successful case studies along with appropriate references should be provided, which would make the presented results more convincing and reliable.

**Response**: This is a helpful point in demonstrating the reliability of the adopted method; thank you for raising it. The SMS-MetaD was chosen for this study for two reasons: *i*) By partitioning the reaction potential energy surface (PES), the SMS-MetaD method allows a more efficient exploration of the free energy landscape of the reaction process, avoiding excessive time being wasted in overly deep wells around the stable minima of reactants and

products; *ii*) The SMS-MetaD is well-suited for effectively modeling chemical systems with complex PES, such as chemical reactions at the air-water interface. Prof. Zhu's group provides more detailed explanations of SMS-MetaD (Fang et al., 2022). This method has already been successfully employed for several theoretical studies, especially investigations into the chemical reaction at the air-water interface (Fang et al., 2024a, b; Tang et al., 2024; Wan et al., 2023). Accordingly, we have added the explanation for selecting SMS-MetaD approach in the Methods section in revised manuscript as follows (Page 4, line 117):

"The SMS-MetaD method is well-suited for effectively modeling chemical systems with complex potential energy surface, especially at the air-water interface (Fang et al., 2022), which has already been successfully employed in the studies of heterogeneous reactions (Fang et al., 2024a, b; Tang et al., 2024; Wan et al., 2023)."
* * *
2. Beyond iodic acid, other iodine oxoacids such as HIO2 and HOI also exist, along with various iodine oxides. In addition, there are many more pollutants, for instance, nitric acid and fluorinated carboxylic acids. Could these species also have an impact? Of course, I am not suggesting that additional calculations be included in this work, but at the very least, the manuscript should address the current limitations of the study and outline potential directions for future improvement.

Response: We thank the reviewer for this insightful comment highlighting the limitations of our manuscript. We agree with the reviewer that other species may also influence this reaction to some extent and should be investigated in future work. The scope of this study does not exclude these possibilities; rather, we focus here on the effects of representative chemical species. A more detailed discussion has been provided in our response to Reviewer 1, and corresponding clarifications have also been added to the revised manuscript (Page 12, Line 304), as follows: "The real atmosphere is chemically complex, including iodine species (e.g. HOI, HIO2, HIO3, I2O3, I2O4, and I2O5) and atmospheric pollutants (e.g. H2SO4, HNO3, organic acids, and ammonia), which are likely to influence the heterogeneous hydrolysis of I2O5. In future work, we intend to confirm the impacts from other atmospheric components."
* * *
3. In Fig. 6, the process of a gas-phase DMA approaching the interfacial I2O5 is presented. However, conversely, would a gas-phase I2O5 approaching the interfacial DMA also lead to a reaction? Why was this scenario not considered?

**Response**: We appreciate the reviewer's keen observation and thoughtful comment, which highlights an important aspect of the reaction dynamics at the interface. Indeed, the mutual approach of the two species could give rise to the two scenarios mentioned by the reviewer. However, since aerosols are generally acidic, the base DMA is unlikely to persist for long and would be protonated to DMAH+, occupying its reactive site and preventing participation in subsequent I2O5 hydrolysis. Therefore, in this study, we only considered the approach of gas-phase DMA toward interfacial I2O5.
* * *
**4.** In the supporting information, some figure annotations or captions should be more detailed. For example, in Fig. S14, the arrows appear to indicate the ESP maxima or minima of the product molecule. Although I can make an educated guess, the authors should provide clearer labels or explanatory notes.

**Response**: Thank reviewer for the perfection of our figures. The explanation of the figure appears only in the caption; there remains a gap between the figure's details and what we intend to convey. We have supplemented the explanatory notes for the arrows, ESP maxima and minima as following,

"The yellow sites indicate" have been changed into "The yellow sites (pointed by yellow arrows) indicate" and "The cyan sites indicate" have been changed into "The cyan sites (pointed by cyan arrows) indicate".

In addition, we have confirmed that the remaining figures are free of similar issues.

**Figure S17.** The electrostatic potential (ESP)-mapped molecular vdW surfaces of the interfacial reaction products mediated by (a) MA, (b) DMA, and (c) TMA. The red regions are electron-deficient, and the blue regions are electron-rich. The yellow sites (pointed by yellow arrows) indicate the points of local ESP maximum; the cyan sites (pointed by cyan arrows) indicate the points of local ESP minimum.
* * *
**5.** For the ELF results shown in Fig. S15, the meaning of the different colored regions should be clarified, as many readers may not be specialists in theoretical studies. For the other figures and captions, I will not list further examples. However, I suggest that the authors carefully re-examine whether the information provided is sufficiently detailed, and consider it from the perspective of a non-specialist reader.

**Response**: We appreciate the reviewer's comment, which enables us to refine our work from the reader's perspective. A corresponding color bar is shown on the right side of each figure, indicating the color scale from high to low (red-green-blue) for the ELF values. We have emphasized the color bar mapping to electron-density magnitude and added explanations of the colored regions in the figure annotation in the Supporting Information as follows:

**Figure S18.** Color-mapped ELF for the iodine products (IO3- and HIO3) of I2O5 hydrolysis mediated by (a) MA, (b) DMA, and (c) TMA. The ELF values (1 to 0) are mapped on a red-green-blue color scale indicated on the right of each subplot.

\_\_\_\_\_

**6.** It may be helpful to revise Scheme 1 to more clearly reflect the key ideas presented in the manuscript. For example, the ionic products that contribute to enhanced hygroscopicity are mentioned in the text but are not explicitly shown in the current scheme. In addition, specifying the molecular pairs responsible for hydrogen and halogen bonding would improve

clarity; iodic acid, for instance, may engage in both hydrogen bonding and halogen bonding with water. The central label of "low volatility" could also be made more explicit by indicating the specific species it refers to.

**Response**: Thanks for the reviewer's valuable comments. The Scheme 1 was intended to overview and summarize the Conclusions section by highlighting several representative species to convey the main idea. Consequently, mechanistic details were largely absent. We have reconsidered the role of the scheme in the manuscript and agree with the reviewer's suggestion; accordingly, we have redrawn Scheme 1 to present the complete mechanism and to include the information missing from the earlier figure.

**Scheme 1.** Illustration of aerosol growth driven by I2O5 hydrolysis at the air-water interface, highlighting the potential reaction pathways and resulting products in pristine and polluted environments.
* * *
7. The mechanism mediated by DMA in Fig. 6 would be better presented in a manner consistent with the others, and I recommend that the corresponding reaction equation be included for completeness.

**Response**: Thank the reviewer for careful attention to the figure details. Indeed, amine-mediated mechanisms (MA, DMA, and TMA) were examined via unbiased BOMD simulations, whereas other species were studied using the SMS-MetaD approach. As these methods provide different types of insights—BOMD and SMS-MetaD simulations focus on the time evolution and free energy changes of key structures in the reaction process,

respectively, which leads to slightly different presentations. To enhance consistency in data presentation, we have added the corresponding reaction equation in Figures 6 and S5-6.
* * *
**Suggested corrections**

**Line 40**: Should "a typical higher I2O2-5" be "one of the highest iodine oxides"?

Line 48: "More recently, the experimental evidence" --> "A most recent experimental evidence"

Line 51: "found the direct" --> "found that the direct"

Line 63: "HIO3 abundant" --> "HIO3 is abundant"

Line 93: Check the word "consisting"

Page 11: Make sure Scheme 1 is clear enough.

**Response**: According to the reviewer's suggestions, we have completed all corresponding revisions.

\_\_\_\_\_

**Reference:**

- Fang, Y.-G., Li, X., Gao, Y., Cui, Y.-H., Francisco, J. S., Zhu, C., and Fang, W.-H.: Efficient exploration of complex free energy landscapes by stepwise multi-subphase space metadynamics, J. Chem. Phys., 157, 214111, https://doi.org/10.1063/5.0098269, 2022.
- Fang, Y.-G., Tang, B., Yuan, C., Wan, Z., Zhao, L., Zhu, S., Francisco, J. S., Zhu, C., and Fang, W.-H.: Mechanistic insight into the competition between interfacial and bulk reactions in microdroplets through N2O5 ammonolysis and hydrolysis, Nat. Commun., 15, 2347, https://doi.org/10.1038/s41467-024-46674-1, 2024a.
- Fang, Y.-G., Wei, L., Francisco, J. S., Zhu, C., and Fang, W.-H.: Mechanistic Insights into Chloric Acid Production by Hydrolysis of Chlorine Trioxide at an Air–Water Interface, J. Am. Chem. Soc., 146, 21052–21060, https://doi.org/10.1021/jacs.4c06269, 2024b.
- Tang, B., Bai, Q., Fang, Y.-G., Francisco, J. S., Zhu, C., and Fang, W.-H.: Mechanistic Insights into N2O5-Halide Ions Chemistry at the Air–Water Interface, J. Am. Chem. Soc., 146, 21742–21751, https://doi.org/10.1021/jacs.4c05850, 2024.
- Wan, Z., Fang, Y., Liu, Z., Francisco, J. S., and Zhu, C.: Mechanistic Insights into the Reactive Uptake of Chlorine Nitrate at the Air–Water Interface, J. Am. Chem. Soc., 145, 944–952, https://doi.org/10.1021/jacs.2c09837, 2023.

---

## Author Comment (AC3)

**Responses to Referee #3's comments**

We are grateful to the reviewers for their professional and helpful comments on our manuscript "Mechanistic Insights into I2O5 Heterogeneous Hydrolysis and Its Role in Iodine Aerosol Growth in Pristine and Polluted Atmospheres" (MS No.: egusphere-2025-3770). Accordingly, we have carefully revised the manuscript. The point-to-point responses to the Referee #3's comments are summarized below:

This manuscript examines how gas-liquid interfacial reactions of higher iodine oxide (I2O5) influence the formation of marine iodine aerosols using molecular dynamics simulations. The authors elucidate the I2O5 heterogeneous hydrolysis mechanism, emphasizing the catalytic effects of atmospheric chemicals. These mechanisms are likely to provide evidence for the extensive presence of iodate in aqueous aerosols, offering guidance for refining atmospheric models of aerosol burden and radiative forcing. As a well-designed theoretical study with atmospheric implications, I recommend this work for publication, subject to my comments being addressed.

**Response**: Thanks for the review's valuable comments and suggestions on our manuscript. The comments have greatly helped us improve the quality of the paper. We have responded to each point carefully and revised the manuscript accordingly.
* * *
The authors suggest that iodine-mediated reactions are more likely to play an important role in pristine environments, whereas pollutant-mediated reactions dominate in polluted marine environments. These heterogeneous mechanisms may have significant atmospheric impacts and need to be evaluated by embedding them into atmospheric models. What challenges do the authors foresee in implementing this cross-scale simulation?

Response: The reviewer's valuable comment has prompted us to consider how to effectively connect microscopic mechanisms with macroscopic environmental impacts. For heterogeneous reaction kinetics, obtaining reaction rates under different environmental conditions remains highly challenging, because the concentrations of I2O5 in the air or aerosols are largely uncertain. Moreover, the coupled effects of temperature, humidity, aerosol size, pH, aging, ions, and other components remain unknown. For atmospheric modeling, it is challenging to obtain a reliable and comprehensive emission inventory and,

based on this, to construct a simulation of the physicochemical transformations from source species to the reactive components of interest. This is followed by the additional difficulty of calibrating the simulated concentrations against field measurements. In summary, the lack of data on heterogeneous reaction kinetics and atmospheric modeling renders cross-scale studies linking interfacial reaction mechanisms to environmental impacts highly challenging.
* * *
**Line 11:** I am not quite sure whether the expression 'higher iodine oxides' is widely used and easily understood, and the authors should check this.

**Response**: The reviewer's comment is quite thorough. In fact, this expression, referring to I2O3-5, appears in many iodine-related studies (Huang et al., 2022; Kaltsoyannis and Plane, 2008; Lewis et al., 2020; Ning et al., 2024; Pound et al., 2024). Many researchers favor higher iodine oxides, while others prefer 'higher-order' (Gómez Martín et al., 2022; Lewis et al., 2020). Considering that 'higher-order' better represents the higher oxidation state of I, we ultimately chose 'higher-order iodine oxides'. We have changed 'higher iodine oxides' into 'higher-order iodine oxides' in the revised manuscript (Page 1, line 11; page 11, line 277).
* * *
**Line 20:** I suggest that the authors moderate some of their conclusions, as this study does not quantify the rates of the relevant chemical processes. Expressions such as "highly effective" should therefore be toned down. More generally, conclusions should avoid absolute or overly strong wording unless supported by quantitative data.

**Response**: The reviewer's suggestion is crucial for improving the rigor of the summary and abstract. As the theoretical findings in this manuscript focus on reaction mechanisms rather than quantitative rate information, terms such as "highly" are inappropriate. we have softened the overstrong statements in the manuscript, with the specific changes detailed as follows: we have changed "highly effective" into "relatively effective" (Page 1, line 20) and "a critical step" into "an unheeded step" (Page 1, line 22).
* * *
Line 31: The reference to Barnes et al. lacks bibliographic details—specifically the year of publication—and similar issues should be checked throughout the reference list. In addition, many studies have examined DMS-derived sulfur and its relation to aerosols; citing only two

papers is insufficient. Please include more primary studies and relevant reviews to support the claim.

**Response**: We appreciate the reviewer's careful review. We have supplemented the bibliographic details of the reference to Barnes et al., and have confirmed the reference list. As suggested by the reviewer, historically, extensive studies have investigated DMS-driven sulfur, particularly the role of methanesulfonic acid (MSA) in aerosol nucleation; thus, the current citations are insufficient to support this point. To address this, we have supplemented more references to enrich the studies for DMS-derived sulfur and its relation to aerosols (Li et al., 2024; Ning and Zhang, 2022; Shen et al., 2020; Zhang et al., 2022, 2024).
* * *
**Line 38:** In the section addressing the uncertain fate of  $I_xO_y$ , the authors should consider citing the experimental study by Finkenzeller et al. (*Nat. Chem.*, 2023, 15, 129). They argued that stable  $I_2O_3$  should be observable, but it was not, which also highlights the uncertainty in the fate of  $I_xO_y$  and supports the view of an unclear  $I_xO_y$  sink.

**Response**: We admire the reviewer's thorough knowledge of the field. The study by Finkenzeller et al. (2023) provides a complete evolution pathway for iodine oxides. Accordingly, we have supplemented this reference in the introduction as suggested.

\_\_\_\_\_

**Line 44:** "...particles through the reaction (R1:  $2HIO_3 \rightarrow I_2O_5 + H_2O$ )" should be revised to the more accurate wording "...particles through the dehydration reaction (R1:  $2HIO_3 \rightarrow I_2O_5 + H_2O$ )."

**Response**: According to the reviewer's suggestion, the sentence "...particles through the reaction (R1:  $2\text{HIO}_3 \rightarrow I_2\text{O}_5 + H_2\text{O}$ )" has been corrected to "...particles through the dehydration reaction (R1:  $2\text{HIO}_3 \rightarrow I_2\text{O}_5 + H_2\text{O}$ )" in the revised manuscript (Page 2, line 44).

\_\_\_\_\_

**Line 64:** Iodate, rather than HIO3, is likely abundant in the aerosol; therefore, the statement should be revised to 'HIO3 (detected as IO3-)'."

**Response**: Line 64 "HIO3" is revised to "HIO3 (detected as IO3-)".
* * *
Line 262: In Scheme 1, according to the caption, the figure is supposed to illustrate the mechanism. However, it actually lacks many details and resembles more of a TOC-style figure. I suggest that the authors replace it with a figure that presents a more detailed depiction of the heterogeneous mechanism.

**Response:** We appreciate the reviewer's constructive comments. We have provided a detailed explanation of this issue in our response to Reviewer 2 and included the revised figure. The latest version of the figure has been updated in the revised manuscript.

**Scheme 1.** Illustration of aerosol growth driven by I2O5 hydrolysis at the air-water interface, highlighting the potential reaction pathways and resulting products in pristine and polluted environments.
* * *
**Reference:**

- Finkenzeller, H., Iyer, S., He, X.-C., Simon, M., Koenig, T. K., Lee, C. F., Valiev, R., Hofbauer, V., Amorim, A., Baalbaki, R., Baccarini, A., Beck, L., Bell, D. M., Caudillo, L., Chen, D., Chiu, R., Chu, B., Dada, L., Duplissy, J., Heinritzi, M., Kemppainen, D., Kim, C., Krechmer, J., Kürten, A., Kvashnin, A., Lamkaddam, H., Lee, C. P., Lehtipalo, K., Li, Z., Makhmutov, V., Manninen, H. E., Marie, G., Marten, R., Mauldin, R. L., Mentler, B., Müller, T., Petäjä, T., Philippov, M., Ranjithkumar, A., Rörup, B., Shen, J., Stolzenburg, D., Tauber, C., Tham, Y. J., Tomé, A., Vazquez-Pufleau, M., Wagner, A. C., Wang, D. S., Wang, M., Wang, Y., Weber, S. K., Nie, W., Wu, Y., Xiao, M., Ye, Q., Zauner-Wieczorek, M., Hansel, A., Baltensperger, U., Brioude, J., Curtius, J., Donahue, N. M., Haddad, I. E., Flagan, R. C., Kulmala, M., Kirkby, J., Sipilä, M., Worsnop, D. R., Kurten, T., Rissanen, M., and Volkamer, R.: The gas-phase formation mechanism of iodic acid as an Chem., atmospheric aerosol Nat. 15. 129 - 135, source, https://doi.org/10.1038/s41557-022-01067-z, 2023.
- Gómez Martín, J. C., Lewis, T. R., James, A. D., Saiz-Lopez, A., and Plane, J. M. C.: Insights into the Chemistry of Iodine New Particle Formation: The Role of Iodine Oxides and the Source of Iodic Acid, J. Am. Chem. Soc., 144, 9240–9253, https://doi.org/10.1021/jacs.1c12957, 2022.
- Huang, R.-J., Hoffmann, T., Ovadnevaite, J., Laaksonen, A., Kokkola, H., Xu, W., Xu, W., Ceburnis, D., Zhang, R., Seinfeld, J. H., and O'Dowd, C.: Heterogeneous iodine-organic chemistry fast-tracks marine new particle formation, Proc. Natl. Acad. Sci., 119, e2201729119, https://doi.org/10.1073/pnas.2201729119, 2022.
- Kaltsoyannis, N. and Plane, J. M. C.: Quantum chemical calculations on a selection of iodine-containing species (IO, OIO, INO3, (IO)2, I2O3, I2O4 and I2O5) of importance in the atmosphere, Phys. Chem. Chem. Phys., 10, 1723, https://doi.org/10.1039/b715687c, 2008.
- Lewis, T. R., Gómez Martín, J. C., Blitz, M. A., Cuevas, C. A., Plane, J. M. C., and Saiz-Lopez, A.: Determination of the absorption cross sections of higher-order iodine oxides at 355 and 532 nm, Atmos. Chem. Phys., 20, 10865–10887, https://doi.org/10.5194/acp-20-10865-2020, 2020.
- Li, J., Wu, N., Chu, B., Ning, A., and Zhang, X.: Molecular-level study on the role of methanesulfonic acid in iodine oxoacid nucleation, Atmos. Chem. Phys., 24, 3989–4000, https://doi.org/10.5194/acp-24-3989-2024, 2024.
- Ning, A. and Zhang, X.: The synergistic effects of methanesulfonic acid (MSA) and methanesulfinic acid (MSIA) on marine new particle formation, Atmos. Environ., 269, 118826, https://doi.org/10.1016/j.atmosenv.2021.118826, 2022.
- Ning, A., Li, J., Du, L., Yang, X., Liu, J., Yang, Z., Zhong, J., Saiz-Lopez, A., Liu, L., Francisco, J. S., and Zhang, X.: Heterogenous Chemistry of I2O3 as a Critical Step in Iodine Cycling, J. Am. Chem. Soc., 146, 33229–33238, https://doi.org/10.1021/jacs.4c13060, 2024.
- Pound, R. J., Brown, L. V., Evans, M. J., and Carpenter, L. J.: An improved estimate of inorganic iodine emissions from the ocean using a coupled surface microlayer box model, Atmos. Chem. Phys., 24, 9899–9921,

- https://doi.org/10.5194/acp-24-9899-2024, 2024.
- Shen, J., Elm, J., Xie, H.-B., Chen, J., Niu, J., and Vehkamäki, H.: Structural Effects of Amines in Enhancing Methanesulfonic Acid-Driven New Particle Formation, Environ. Sci. Technol., 54, 13498–13508, https://doi.org/10.1021/acs.est.0c05358, 2020.
- Zhang, R., Shen, J., Xie, H.-B., Chen, J., and Elm, J.: The role of organic acids in new particle formation from methanesulfonic acid and methylamine, Atmos. Chem. Phys., 22, 2639–2650, https://doi.org/10.5194/acp-22-2639-2022, 2022.
- Zhang, R., Ma, F., Zhang, Y., Chen, J., Elm, J., He, X.-C., and Xie, H.-B.: HIO3–HIO2 -Driven Three-Component Nucleation: Screening Model and Cluster Formation Mechanism, Environ. Sci. Technol., 58, 649–659, https://doi.org/10.1021/acs.est.3c06098, 2024.

---

## Author Response (AR1)

**Responses to Referee #1's comments**

We are grateful to the reviewers for their professional and helpful comments on our manuscript "**Mechanistic Insights into $I_2O_5$ Heterogeneous Hydrolysis and Its Role in Iodine Aerosol Growth in Pristine and Polluted Atmospheres**" (MS No.: egusphere-2025-3770). Accordingly, we have carefully revised the manuscript. The point-to-point responses to the Referee #1's comments are summarized below:

Deng et al. presented a theoretical study showing that, under the influence of atmospheric iodine species and pollutants, $I_2O_5$ hydrolysis can occur more readily at the surface of aqueous aerosols. These physicochemical processes are valuable, as $I_2O_5$, being a key chemical in the iodine cycling, has a significant impact on both iodine chemistry and the formation of iodine aerosols. Experimental investigation of gas−liquid interfacial reactions is challenging; therefore, the heterogeneous mechanisms revealed by the authors through Ab initio molecular dynamics simulations provide an important advancement of the previous understanding of the atmospheric fate of iodine oxides. This manuscript is thoughtfully prepared, with reliable methods and comprehensive data in both the main text and the supplementary material that support the conclusions. That said, certain aspects could benefit from minor revision, and I recommend publication after the authors have addressed my comments.

**Response:** Thanks for the reviewer's professional and valuable comments. We have addressed all comments point by point and made the corresponding revisions in the manuscript. The detailed responses are listed as follows.
* * *
**Major Comment**:

**Page 3, lines 84-85**: It is stated that gas-phase structure optimizations were performed with Gaussian package, yet the species involved are not clearly identified. As I could not find this information in either the manuscript or the SI. If I have overlooked it, please direct me to the relevant section.

**Response**: We thank the reviewer for the careful review. This suggestion has reminded us of the missing details in the manuscript. In this study, we optimized the gas-phase conformations of the reactants (e.g., $I_2O_5$, $HIO_3$, and $H_2SO_4$) using the Gaussian 16 program (Frisch et al.,

2016). Accordingly, we have added the relevant molecular information in the Methods section in the revised manuscript (Page 3, line 92), as follows:

"The geometries and coordinates of gas-phase molecules (i.e. $I_2O_5$, $HIO_3$, $H_2SO_4$, MA, DMA, and TMA) are provided in Figure S2 and Table S2 in the supporting information (SI), respectively." in the Quantum Chemistry Calculations section.

For ease of review, we have copied them as follows:

[Figure]

**Figure S2.** The optimized structures for gas-phase molecules (i.e. $I_2O_5$, $HIO_3$, $H_2SO_4$, methylamine (MA), dimethylamine (DMA), and trimethylamine (TMA)) at the M06-2X//aug-cc-pVTZ(-PP) level of theory.

**Table S2.** Coordinates for all calculated molecules at the M06-2X/aug-cc-pVTZ(-PP) level of theory.

| $I_2O_5$ (Isomer 1) | | | |
|---|---|---|---|
| O | -0.000013 | -0.988788 | 0.000159 |
| I | 1.641204 | 0.055287 | -0.225857 |
| I | -1.641235 | 0.055311 | 0.225852 |
| O | 2.800249 | -1.108887 | 0.404898 |
| O | 1.270721 | 1.236952 | 1.040258 |
| O | -2.800236 | -1.108859 | -0.404989 |
| O | -1.270515 | 1.236874 | -1.040296 |

**HIO₃**

| | | | |
|---|---|---|---|
| I | -0.096935 | 0.013334 | -0.244580 |
| O | -0.289743 | 1.559142 | 0.594271 |
| O | -1.014657 | -1.204758 | 0.647080 |
| O | 1.713186 | -0.419126 | 0.233680 |
| H | 1.867280 | -0.188774 | 1.162521 |

**H₂SO₄**

| | | | |
|---|---|---|---|
| S | 0.000001 | -0.000003 | 0.154681 |
| O | 0.670149 | 1.064615 | 0.826194 |
| O | 1.023747 | -0.686964 | -0.845067 |
| H | 1.708834 | -0.047610 | -1.086390 |
| O | -1.023744 | 0.686990 | -0.845052 |
| H | -1.708821 | 0.047636 | -1.086403 |
| O | -0.670155 | -1.064638 | 0.826162 |

**MA**

| | | | |
|---|---|---|---|
| N | 0.747530 | 0.000000 | -0.120845 |
| H | 1.149839 | -0.811832 | 0.328231 |
| H | 1.149839 | 0.811832 | 0.328231 |
| C | -0.706161 | 0.000000 | 0.017785 |
| H | -1.112738 | 0.876800 | -0.483829 |
| H | -1.112739 | -0.876799 | -0.483831 |
| H | -1.069944 | -0.000001 | 1.050407 |

**DMA**

| | | | |
|---|---|---|---|
| N | 0.000000 | 0.568475 | -0.148304 |

| | | | |
|---|---|---|---|
| H | 0.000001 | 1.336588 | 0.508868 |
| C | -1.204665 | -0.224024 | 0.020309 |
| H | -2.083499 | 0.413833 | -0.056533 |
| H | -1.258202 | -0.965087 | -0.778566 |
| H | -1.244160 | -0.762560 | 0.977876 |
| C | 1.204665 | -0.224024 | 0.020309 |
| H | 1.258206 | -0.965080 | -0.778573 |
| H | 2.083499 | 0.413834 | -0.056523 |
| H | 1.244156 | -0.762568 | 0.977871 |

TMA

| | | | |
|---|---|---|---|
| N | 0.000000 | 0.000009 | -0.389205 |
| C | 1.195212 | 0.683133 | 0.062542 |
| H | 2.079355 | 0.170403 | -0.315039 |
| H | 1.202277 | 1.705136 | -0.315089 |
| H | 1.260883 | 0.720681 | 1.163047 |
| C | -1.189213 | 0.693506 | 0.062541 |
| H | -1.187224 | 1.715614 | -0.314877 |
| H | -2.077799 | 0.188704 | -0.315260 |
| H | -1.254710 | 0.731401 | 1.163041 |
| C | -0.005990 | -1.376629 | 0.062538 |
| H | -0.892218 | -1.885891 | -0.314819 |
| H | 0.875409 | -1.893854 | -0.315333 |
| H | -0.006029 | -1.452314 | 1.163040 |
* * *
For the central chemical examined in this work, the $I_2O_5$ molecule, different isomers are expected to exist. Could the authors clarify why the current structure was selected and on what basis? Moreover, in the introduction the $I_2O_5$ molecule is described as very stable; does the cited reference pertain to the same structure investigated here? The rationale for the chosen structure should be stated, and the atomic coordinates together with a structural figure are best included in the SI.

**Response**: The reviewer's comment is professional. In our manuscript, the isomers of $I_2O_5$ had been already considered before carrying out BOMD simulations. We have surveyed the previously reported structures of $I_2O_5$ (Kaltsoyannis and Plane, 2008; Khanniche et al., 2016; Kim and Yoo, 2016). Although isomer 2 (Kaltsoyannis and Plane, 2008) is referred to in the Introduction as a stable configuration, our calculations show that isomer 1 (Khanniche et al., 2016) is in fact more stable, with a lower Gibbs free energy. Thus, the most stable isomer 1 was selected for the subsequent BOMD simulations. We have supplemented description for isomers of $I_2O_5$ in the Methods section as follows (Page 3, line 86):

"The $I_2O_5$ molecule with lowest Gibbs free energy has been selected from isomers (Kaltsoyannis and Plane, 2008; Khanniche et al., 2016; Kim and Yoo, 2016) , and details of the structures and coordinates are provided in SI (Figure S1 and Table S3)."

To ensure clarity for the readers, in the revised SI, we have presented the considered isomers together with their calculated Gibbs free energies in Fig. S1, and the corresponding coordinates are summarized in Table S3.

[Figure]

| $\Delta G$ (kcal/mol) | Isomer 1 | Isomer 2 | Isomer 3 |
|---|---|---|---|
| | 0.00 | 0.71 | 3.92 |

**Figure S1.** The optimized structures for isomers of $I_2O_5$ at the M06-2X/aug-cc-pVTZ(-PP) level of theory. The relative Gibbs free energies (kcal/mol, comparing to isomer 1) are provided beneath the corresponding isomers.

**Table S3.** Coordinates for all calculated isomers of $I_2O_5$ at the M06-2X/aug-cc-pVTZ(-PP) level of theory.

| | | | |
|---|---|---|---|
| | | Isomer 1 | |
| O | -0.000013 | -0.988788 | 0.000159 |
| I | 1.641204 | 0.055287 | -0.225857 |
| I | -1.641235 | 0.055311 | 0.225852 |
| O | 2.800249 | -1.108887 | 0.404898 |
| O | 1.270721 | 1.236952 | 1.040258 |
| O | -2.800236 | -1.108859 | -0.404989 |
| O | -1.270515 | 1.236874 | -1.040296 |
| | | Isomer 2 | |
| I | 0.000000 | 1.656872 | -0.054736 |
| O | 0.000000 | 0.000000 | 0.989073 |
| I | 0.000000 | -1.656872 | -0.054736 |
| O | -0.789433 | -2.716528 | 1.107431 |
| O | 1.199506 | 1.113881 | -1.239338 |
| O | 0.789433 | 2.716528 | 1.107431 |
| O | -1.199506 | -1.113881 | -1.239338 |
| | | Isomer 3 | |
| I | -1.769146 | -0.160199 | -0.157389 |
| O | -0.000002 | 0.000007 | -0.980462 |
| I | 1.769147 | 0.160192 | -0.157396 |
| O | -2.140622 | 1.528856 | 0.182916 |
| O | -1.400607 | -0.996020 | 1.350040 |
| O | 1.400657 | 0.996070 | 1.350017 |

|   |   |   |   |
|---|---|---|---|
| O | 2.140569 | -1.528872 | 0.182940 |
* * *
Across the heterogeneous hydrolysis pathways of $I_2O_5$ presented in this study, whether mediated by water, iodic acid, $I_2O_5$, or pollutants, the cleavage always occurs at the central I–O covalent bond of the $I_2O_5$. This appears to be a consequence of the CV definition, which biases the system toward iodic acid formation. Nevertheless, the rationale for this setup should be substantiated by chemical evidence. A wavefunction analysis, such as bond order calculations, could be provided to confirm that the central I–O bond is indeed the weak, thereby justifying its designation as the most likely bond to break.

**Response**: According to the reviewer's suggestion, we have calculated the Mayer bond order (MBO) of adopted $I_2O_5$ molecule by Multiwfn 3.7 (Lu and Chen, 2012). As shown in Figure S3, the central I-O bond is considered to be a single bond (MBO: 0.864), while the terminal I-O bond is thought to be a double bond (MBO: 1.729). The results indicate that initial cleavage is expected to occur at the central but weaker I-O covalent bond of the $I_2O_5$. We have supplemented the chemical-bond characterization for this part in the SI as the reviewer suggested. This result provides compelling chemical evidence supporting the rationality of the CV settings.

[Figure]

**Figure S3.** Mayer bond orders for $I_2O_5$ calculated at the M06-2X/cc-pVTZ(-PP) level of theory.
* * *
Atmospheric iodine species and pollutants are more diverse than the limited set examined here. For example, even in the case of amines, more than one hundred species exist in the atmosphere. It would be helpful if the authors could include a brief discussion of the possible roles of other atmospheric components, or at minimum acknowledge this as a limitation of the current study.

**Response**: Thank you for this insightful suggestion. This helps readers understand the limitations of the study. The real atmosphere is complex; as the reviewer noted, iodine species (e.g. HOI, $HIO_2$, $HIO_3$, $I_2O_3$, $I_2O_4$, and $I_2O_5$) and atmospheric pollutants (e.g. $H_2SO_4$, $HNO_3$, dimethyl sulfide, organic acids, and aromatic hydrocarbons)-including amines-are highly diverse. We consider that other components are also likely to influence the heterogeneous hydrolysis process of the $I_2O_5$ of interest. In this study, the effects of $I_2O_5$ and $HIO_3$ on the reaction are explored here mainly because, as reactants and products, they are most likely to coexist in the same environment, thereby facilitating self catalysis. Meanwhile, we chose $H_2SO_4$ and amines (i.e. MA, DMA, and TMA) as the representative acid and base pollutants that are associated with aerosol particle formation. We have expanded this part of the discussion to better reflect real atmospheric conditions and acknowledge the limitation that the manuscript can not comprehensively examine all species in revised manuscript as follows (Page 12, line 304):

"The real atmosphere is chemically complex, including iodine species (e.g. HOI, $HIO_2$, $HIO_3$, $I_2O_3$, $I_2O_4$, and $I_2O_5$) and atmospheric pollutants (e.g. $H_2SO_4$, $HNO_3$, organic acids, and ammonia), which are likely to influence the heterogeneous hydrolysis of $I_2O_5$. In future work, we intend to confirm the impacts from other atmospheric components."
* * *
**Minor Comments:**

**Page 5, Line 139**: "…along the Z axis…" Units are missing.

**Page6, Line 165**: "Pink, red, white atoms represent I, O, H in sequence (The same below)." This sentence should be: "The pink, red, and white spheres represent I, O, and H atoms, respectively (the same applies in Figures 3–6 below)."

**Lines 203 and 226**: 'Profiles' should be 'The profile'

**Lines 188**: 'error bands' should be 'error band'

**Line 240**: It is recommended to remove this citation, as it does not appear to provide effective support.

**Response**: We appreciate the reviewer's careful evaluation. Accordingly, we have completed all corresponding revisions in response to the reviewer's minor comments.
* * *
**Responses to Referee #2's comments**

We are grateful to the reviewers for their professional and helpful comments on our manuscript "**Mechanistic Insights into I$_2$O$_5$ Heterogeneous Hydrolysis and Its Role in Iodine Aerosol Growth in Pristine and Polluted Atmospheres**" (MS No.: egusphere-2025-3770). Accordingly, we have carefully revised the manuscript. The point-to-point responses to the Referee #2's comments are summarized below:

The manuscript by Deng et al. adopted first-principles molecular dynamics to examine the heterogeneous hydrolysis of I$_2$O$_5$ and its role in aerosol growth. The study identifies interfacial mechanisms driven by iodic species in pristine conditions and by acid/base pollutants in polluted environments. These findings emphasize the importance of reactive atmospheric components in I$_2$O$_5$ hydrolysis and provide a mechanistic explanation for its sink and the observed I$_2$O$_5$-to-HIO$_3$ conversion. This topic is timely and relevant to Atmos. Chem. Phys., given its focus on aerosol formation process. The manuscript is overall well presented, with sound methodology and adequate supporting evidence. Nevertheless, I suggest that the authors consider my comments and perform the minor revisions before the manuscript can be recommended for publication.

**Response**: We appreciate the insightful and constructive suggestions. According to these comments, we have responded and revised the manuscript as follows.
* * *
1. In conducting the metadynamics simulations, the authors appear to have employed SMS-MetaD rather than the more widely used MetaD approach. I am curious about the rationale behind this choice. The authors should explain the advantages of this method in the Methods section. Furthermore, to substantiate its feasibility, some successful case studies along with appropriate references should be provided, which would make the presented results more convincing and reliable.

**Response**: This is a helpful point in demonstrating the reliability of the adopted method; thank you for raising it. The SMS-MetaD was chosen for this study for two reasons: *i*) By partitioning the reaction potential energy surface (PES), the SMS-MetaD method allows a more efficient exploration of the free energy landscape of the reaction process, avoiding excessive time being wasted in overly deep wells around the stable minima of reactants and

products; *ii*) The SMS-MetaD is well-suited for effectively modeling chemical systems with complex PES, such as chemical reactions at the air–water interface. Prof. Zhu's group provides more detailed explanations of SMS-MetaD (Fang et al., 2022). This method has already been successfully employed for several theoretical studies, especially investigations into the chemical reaction at the air-water interface (Fang et al., 2024a, b; Tang et al., 2024; Wan et al., 2023). Accordingly, we have added the explanation for selecting SMS-MetaD approach in the Methods section in revised manuscript as follows (Page 4, line 117):

"The SMS-MetaD method is well-suited for effectively modeling chemical systems with complex potential energy surface, especially at the air-water interface (Fang et al., 2022), which has already been successfully employed in the studies of heterogeneous reactions (Fang et al., 2024a, b; Tang et al., 2024; Wan et al., 2023)."
* * *
**2.** Beyond iodic acid, other iodine oxoacids such as $HIO_2$ and $HOI$ also exist, along with various iodine oxides. In addition, there are many more pollutants, for instance, nitric acid and fluorinated carboxylic acids. Could these species also have an impact? Of course, I am not suggesting that additional calculations be included in this work, but at the very least, the manuscript should address the current limitations of the study and outline potential directions for future improvement.

**Response**: We thank the reviewer for this insightful comment highlighting the limitations of our manuscript. We agree with the reviewer that other species may also influence this reaction to some extent and should be investigated in future work. The scope of this study does not exclude these possibilities; rather, we focus here on the effects of representative chemical species. A more detailed discussion has been provided in our response to Reviewer 1, and corresponding clarifications have also been added to the revised manuscript (Page 12, Line 304), as follows: "The real atmosphere is chemically complex, including iodine species (e.g. $HOI$, $HIO_2$, $HIO_3$, $I_2O_3$, $I_2O_4$, and $I_2O_5$) and atmospheric pollutants (e.g. $H_2SO_4$, $HNO_3$, organic acids, and ammonia), which are likely to influence the heterogeneous hydrolysis of $I_2O_5$. In future work, we intend to confirm the impacts from other atmospheric components."
* * *
**3.** In Fig. 6, the process of a gas-phase DMA approaching the interfacial $I_2O_5$ is presented. However, conversely, would a gas-phase $I_2O_5$ approaching the interfacial DMA also lead to a reaction? Why was this scenario not considered?

**Response**: We appreciate the reviewer's keen observation and thoughtful comment, which highlights an important aspect of the reaction dynamics at the interface. Indeed, the mutual approach of the two species could give rise to the two scenarios mentioned by the reviewer. However, since aerosols are generally acidic, the base DMA is unlikely to persist for long and would be protonated to DMAH$^+$, occupying its reactive site and preventing participation in subsequent $I_2O_5$ hydrolysis. Therefore, in this study, we only considered the approach of gas-phase DMA toward interfacial $I_2O_5$.
* * *
**4.** In the supporting information, some figure annotations or captions should be more detailed. For example, in Fig. S14, the arrows appear to indicate the ESP maxima or minima of the product molecule. Although I can make an educated guess, the authors should provide clearer labels or explanatory notes.

**Response**: Thank reviewer for the perfection of our figures. The explanation of the figure appears only in the caption; there remains a gap between the figure's details and what we intend to convey. We have supplemented the explanatory notes for the arrows, ESP maxima and minima as following,

"The yellow sites indicate" have been changed into "The yellow sites (pointed by yellow arrows) indicate" and "The cyan sites indicate" have been changed into "The cyan sites (pointed by cyan arrows) indicate".

In addition, we have confirmed that the remaining figures are free of similar issues.

[Figure]

(a) $HIO_3 + IO_3^- + MAH^+$  (b) $HIO_3 + IO_3^- + DMAH^+$  (c) $HIO_3 + IO_3^- + TMAH^+$

**Figure S17.** The electrostatic potential (ESP)-mapped molecular vdW surfaces of the interfacial reaction products mediated by (a) MA, (b) DMA, and (c) TMA. The red regions are electron-deficient, and the blue regions are electron-rich. The yellow sites (pointed by yellow arrows) indicate the points of local ESP maximum; the cyan sites (pointed by cyan arrows) indicate the points of local ESP minimum.
* * *
**5.** For the ELF results shown in Fig. S15, the meaning of the different colored regions should be clarified, as many readers may not be specialists in theoretical studies. For the other figures and captions, I will not list further examples. However, I suggest that the authors carefully re-examine whether the information provided is sufficiently detailed, and consider it from the perspective of a non-specialist reader.

**Response**: We appreciate the reviewer's comment, which enables us to refine our work from the reader's perspective. A corresponding color bar is shown on the right side of each figure, indicating the color scale from high to low (red-green-blue) for the ELF values. We have emphasized the color bar mapping to electron-density magnitude and added explanations of the colored regions in the figure annotation in the Supporting Information as follows:

[Figure]

**Figure S18.** Color-mapped ELF for the iodine products ($IO_3^-$ and $HIO_3$) of $I_2O_5$ hydrolysis mediated by (a) MA, (b) DMA, and (c) TMA. The ELF values (1 to 0) are mapped on a red-green-blue color scale indicated on the right of each subplot.
* * *
**6.** It may be helpful to revise Scheme 1 to more clearly reflect the key ideas presented in the manuscript. For example, the ionic products that contribute to enhanced hygroscopicity are mentioned in the text but are not explicitly shown in the current scheme. In addition, specifying the molecular pairs responsible for hydrogen and halogen bonding would improve

clarity; iodic acid, for instance, may engage in both hydrogen bonding and halogen bonding with water. The central label of "low volatility" could also be made more explicit by indicating the specific species it refers to.

**Response**: Thanks for the reviewer's valuable comments. The Scheme 1 was intended to overview and summarize the Conclusions section by highlighting several representative species to convey the main idea. Consequently, mechanistic details were largely absent. We have reconsidered the role of the scheme in the manuscript and agree with the reviewer's suggestion; accordingly, we have redrawn Scheme 1 to present the complete mechanism and to include the information missing from the earlier figure.

[Figure]

**Scheme 1.** Illustration of aerosol growth driven by $I_2O_5$ hydrolysis at the air-water interface, highlighting the potential reaction pathways and resulting products in pristine and polluted environments.
* * *
**7.** The mechanism mediated by DMA in Fig. 6 would be better presented in a manner consistent with the others, and I recommend that the corresponding reaction equation be included for completeness.

**Response**: Thank the reviewer for careful attention to the figure details. Indeed, amine-mediated mechanisms (MA, DMA, and TMA) were examined via unbiased BOMD simulations, whereas other species were studied using the SMS-MetaD approach. As these methods provide different types of insights—BOMD and SMS-MetaD simulations focus on the time evolution and free energy changes of key structures in the reaction process,

respectively, which leads to slightly different presentations. To enhance consistency in data presentation, we have added the corresponding reaction equation in Figures 6 and S5-6.
* * *
**Suggested corrections**

**Line 40**: Should "a typical higher $I_2O_{2-5}$" be "one of the highest iodine oxides"?

**Line 48**: "More recently, the experimental evidence" --> "A most recent experimental evidence"

**Line 51**: "found the direct" --> "found that the direct"

**Line 63**: "$HIO_3$ abundant" --> "$HIO_3$ is abundant"

**Line 93**: Check the word "consisting"

**Page 11**: Make sure Scheme 1 is clear enough.

**Response**: According to the reviewer's suggestions, we have completed all corresponding revisions.
* * *
**Responses to Referee #3's comments**

We are grateful to the reviewers for their professional and helpful comments on our manuscript "**Mechanistic Insights into I$_2$O$_5$ Heterogeneous Hydrolysis and Its Role in Iodine Aerosol Growth in Pristine and Polluted Atmospheres**" (MS No.: egusphere-2025-3770). Accordingly, we have carefully revised the manuscript. The point-to-point responses to the Referee #3's comments are summarized below:

This manuscript examines how gas–liquid interfacial reactions of higher iodine oxide (I$_2$O$_5$) influence the formation of marine iodine aerosols using molecular dynamics simulations. The authors elucidate the I$_2$O$_5$ heterogeneous hydrolysis mechanism, emphasizing the catalytic effects of atmospheric chemicals. These mechanisms are likely to provide evidence for the extensive presence of iodate in aqueous aerosols, offering guidance for refining atmospheric models of aerosol burden and radiative forcing. As a well-designed theoretical study with atmospheric implications, I recommend this work for publication, subject to my comments being addressed.

**Response**: Thanks for the review's valuable comments and suggestions on our manuscript. The comments have greatly helped us improve the quality of the paper. We have responded to each point carefully and revised the manuscript accordingly.
* * *
The authors suggest that iodine-mediated reactions are more likely to play an important role in pristine environments, whereas pollutant-mediated reactions dominate in polluted marine environments. These heterogeneous mechanisms may have significant atmospheric impacts and need to be evaluated by embedding them into atmospheric models. What challenges do the authors foresee in implementing this cross-scale simulation?

**Response**: The reviewer's valuable comment has prompted us to consider how to effectively connect microscopic mechanisms with macroscopic environmental impacts. For heterogeneous reaction kinetics, obtaining reaction rates under different environmental conditions remains highly challenging, because the concentrations of I$_2$O$_5$ in the air or aerosols are largely uncertain. Moreover, the coupled effects of temperature, humidity, aerosol size, pH, aging, ions, and other components remain unknown. For atmospheric modeling, it is challenging to obtain a reliable and comprehensive emission inventory and,

based on this, to construct a simulation of the physicochemical transformations from source species to the reactive components of interest. This is followed by the additional difficulty of calibrating the simulated concentrations against field measurements. In summary, the lack of data on heterogeneous reaction kinetics and atmospheric modeling renders cross-scale studies linking interfacial reaction mechanisms to environmental impacts highly challenging.
* * *
**Line 11:** I am not quite sure whether the expression 'higher iodine oxides' is widely used and easily understood, and the authors should check this.

**Response**: The reviewer's comment is quite thorough. In fact, this expression, referring to $I_2O_{3-5}$, appears in many iodine-related studies (Huang et al., 2022; Kaltsoyannis and Plane, 2008; Lewis et al., 2020; Ning et al., 2024; Pound et al., 2024). Many researchers favor higher iodine oxides, while others prefer 'higher-order' (Gómez Martín et al., 2022; Lewis et al., 2020). Considering that 'higher-order' better represents the higher oxidation state of I, we ultimately chose 'higher-order iodine oxides'. We have changed 'higher iodine oxides' into 'higher-order iodine oxides' in the revised manuscript (Page 1, line 11; page 11, line 277).
* * *
**Line 20:** I suggest that the authors moderate some of their conclusions, as this study does not quantify the rates of the relevant chemical processes. Expressions such as "highly effective" should therefore be toned down. More generally, conclusions should avoid absolute or overly strong wording unless supported by quantitative data.

**Response**: The reviewer's suggestion is crucial for improving the rigor of the summary and abstract. As the theoretical findings in this manuscript focus on reaction mechanisms rather than quantitative rate information, terms such as "highly" are inappropriate. we have softened the overstrong statements in the manuscript, with the specific changes detailed as follows: we have changed "highly effective" into "relatively effective" (Page 1, line 20) and "a critical step" into "an unheeded step" (Page 1, line 22).
* * *
**Line 31:** The reference to Barnes et al. lacks bibliographic details—specifically the year of publication—and similar issues should be checked throughout the reference list. In addition, many studies have examined DMS-derived sulfur and its relation to aerosols; citing only two

papers is insufficient. Please include more primary studies and relevant reviews to support the claim.

**Response**: We appreciate the reviewer's careful review. We have supplemented the bibliographic details of the reference to Barnes et al., and have confirmed the reference list. As suggested by the reviewer, historically, extensive studies have investigated DMS-driven sulfur, particularly the role of methanesulfonic acid (MSA) in aerosol nucleation; thus, the current citations are insufficient to support this point. To address this, we have supplemented more references to enrich the studies for DMS-derived sulfur and its relation to aerosols (Li et al., 2024; Ning and Zhang, 2022; Shen et al., 2020; Zhang et al., 2022, 2024).
* * *
**Line 38:** In the section addressing the uncertain fate of $I_xO_y$, the authors should consider citing the experimental study by Finkenzeller et al. (*Nat. Chem.*, 2023, 15, 129). They argued that stable $I_2O_3$ should be observable, but it was not, which also highlights the uncertainty in the fate of $I_xO_y$ and supports the view of an unclear $I_xO_y$ sink.

**Response**: We admire the reviewer's thorough knowledge of the field. The study by Finkenzeller et al. (2023) provides a complete evolution pathway for iodine oxides. Accordingly, we have supplemented this reference in the introduction as suggested.
* * *
**Line 44:** "…particles through the reaction (R1: $2HIO_3 \rightarrow I_2O_5 + H_2O$)" should be revised to the more accurate wording "…particles through the dehydration reaction (R1: $2HIO_3 \rightarrow I_2O_5 + H_2O$)."

**Response**: According to the reviewer's suggestion, the sentence "…particles through the reaction (R1: $2HIO_3 \rightarrow I_2O_5 + H_2O$)" has been corrected to "…particles through the dehydration reaction (R1: $2HIO_3 \rightarrow I_2O_5 + H_2O$)" in the revised manuscript (Page 2, line 44).
* * *
**Line 64:** Iodate, rather than $HIO_3$, is likely abundant in the aerosol; therefore, the statement should be revised to '$HIO_3$ (detected as $IO_3^-$)'."

**Response**: Line 64 "$HIO_3$" is revised to "$HIO_3$ (detected as $IO_3^-$)".
* * *
**Line 262:** In Scheme 1, according to the caption, the figure is supposed to illustrate the mechanism. However, it actually lacks many details and resembles more of a TOC-style figure. I suggest that the authors replace it with a figure that presents a more detailed depiction of the heterogeneous mechanism.

**Response:** We appreciate the reviewer's constructive comments. We have provided a detailed explanation of this issue in our response to Reviewer 2 and included the revised figure. The latest version of the figure has been updated in the revised manuscript.

[revised manuscript text omitted]